# Going Beyond Feature Similarity: Effective Dataset distillation based on Class-aware Conditional Mutual Information

**Xinhao Zhong**[1]    **Bin Chen**[1,2*]    **Hao Fang**[3]    **Xulin Gu**[1]    **Shu-Tao Xia**[3]    **En-Hui Yang**[4]

[1]Harbin Institute of Technology, Shenzhen    [2]Peng Cheng Laboratory
[3]Tsinghua Shenzhen International Graduate School, Tsinghua University    [4]University of Waterloo
`xh021213@gmail.com, fang-h23@mails.tsinghua.edu.cn,`
`chenbin2021@hit.edu.cn, 210110720@stu.hit.edu.cn,`
`ehyang@uwaterloo.ca, xiast@sz.tsinghua.edu.cn;`

## Abstract

Dataset distillation (DD) aims to minimize the time and memory consumption needed for training deep neural networks on large datasets, by creating a smaller synthetic dataset that has similar performance to that of the full real dataset. However, current dataset distillation methods often result in synthetic datasets that are excessively difficult for networks to learn from, due to the compression of a substantial amount of information from the original data through metrics measuring feature similarity, e,g., distribution matching (DM). In this work, we introduce conditional mutual information (CMI) to assess the class-aware complexity of a dataset and propose a novel method by minimizing CMI. Specifically, we minimize the distillation loss while constraining the class-aware complexity of the synthetic dataset by minimizing its empirical CMI from the feature space of pre-trained networks, simultaneously. Conducting on a thorough set of experiments, we show that our method can serve as a general regularization method to existing DD methods and improve the performance and training efficiency. Our code is available at `https://github.com/ndhg1213/CMIDD`.

## 1 Introduction

Dataset distillation (DD) (Wang et al., 2018; Nguyen et al., 2021a;b; Zhou et al., 2022) has received tremendous attention from both academia and industry in recent years as a highly effective data compression technique, and has been deployed in different fields, ranging from continual learning (Gu et al., 2024b), federated learning (Wang et al., 2024b) to privacy-preserving (Dong et al., 2022). Dataset distillation aims to generate a smaller synthetic dataset from a large real dataset, which encapsulates a higher level of concentration of task-specific information than the large real counterpart and which the deep neural networks (DNNs) trained on can attain performance comparable to that of those trained on large real datasets, but with benefits of significant savings in both training time and memory consumption.

Existing dataset distillation methods optimize the smaller synthetic dataset through stochastic gradient descent by matching gradient (Zhao et al., 2021), feature (Zhao & Bilen, 2021a), or statistic information Yin et al. (2024), aiming to compress task-relevant information sensitive to the backbone model into the generated images. Building on this, some recent methods propose using feature clustering (Deng et al., 2024), maximizing mutual information (Shang et al., 2024), or progressive optimization (Chen et al., 2024) to enhance performance. However, these plug-and-play methods focus on improving the alignment of the synthetic dataset with the real dataset in a specific information space, e.g., distribution matching, while neglecting the properties of different classes inherent in the synthetic dataset. This limitation restricts their applicability to specific distillation methods and results in relatively modest performance improvements.

---

*Corresponding Author.

A promising group of methods (Paul et al., 2021; Zheng et al., 2022) on dataset evaluation indicate that in few-shot learning, simpler datasets are often more beneficial for model training. This suggests that while synthetic datasets aim to distillate all the information from the real dataset during optimization, the complexity of distilled information can make it more challenging for models to learn, and the overly complex samples often introduce biases during model training. This limitation has been further supported by recent studies (He et al., 2024; Yang et al., 2024), which show harder samples can not support the entire training and often lead to performance drops and fluctuations.

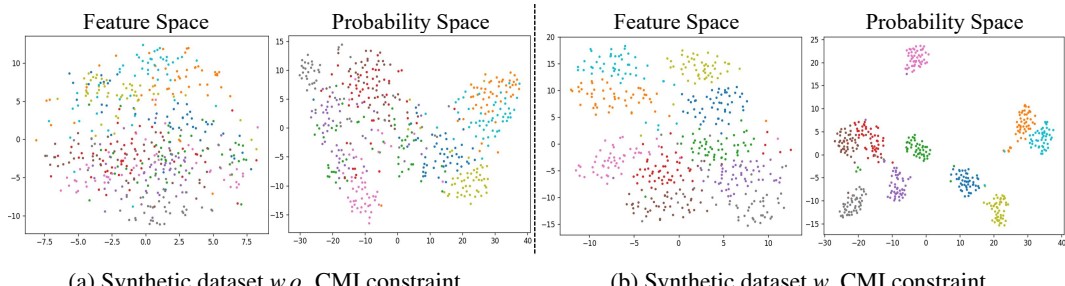

(a) Synthetic dataset *w.o.* CMI constraint.      (b) Synthetic dataset *w.* CMI constraint.

Figure 1: Visualization of the synthetic dataset generated by DM with (a) high CMI value, and (b) low CMI value.

To achieve a more granular measurement of the information involved in the dataset distillation process, especially regarding the varying difficulty levels across different classes, this paper introduces conditional mutual information (CMI) from information theory (Yang et al., 2023; Zhuang et al., 2025) as a class-aware complexity metric for measuring fine-grained condensed information. Specifically, given a pre-trained neural network $f_{\theta^*}(\cdot)$ parameterized by $\theta^*$, we define the CMI $I(S; \hat{Y} \mid Y)$, where $S$, $Y$ and $\hat{Y}$ are three random variables representing the input synthetic dataset, the ground truth label, and the output of $f_{\theta^*}(\cdot)$, respectively. Building on this, $I(S; \hat{Y} \mid Y)$ quantifies the amount of information about $S$ contained in $\hat{Y}$ given the class condition $Y$, indirectly measuring the class-aware complexity of the synthetic dataset by demonstrating how the output of a pre-trained network is influenced by the synthetic dataset. Through empirical verification, we demonstrate that minimizing CMI in the feature space of $f_{\theta^*}(\cdot)$ invariably leads to the distilled data becoming more focused around the center of each class, thereby enhancing the generalization of the distilled data across different network architectures as shown in Figure 1.

Based on this intuition, we propose a novel regularization method to existing DD methods by simultaneously minimizing the distillation loss of the synthetic datasets and its CMI. First, we employ an efficient CMI estimation method to measuring the class-aware inherent properties of the synthetic dataset. Under the guidance of this metric, we treat CMI as a universal regularization method and combine it with existing dataset distillation techniques with diverse optimization objectives. Experiments demonstrate that the CMI enhanced losses significantly outperform various state-of-the-art methods in most cases. Even huge improvements of over 5% can be obtained under some conditions.

In summary, the contributions of the paper are as follows:

- We provide an insight into the properties of different classes inherent in the synthetic dataset and point out that the generalization of the distilled data can be improved by optimizing the class-aware complexity of synthetic dataset quantified via CMI empirically and theoretically.

- Building on this perspective, we propose the CMI enhanced loss that simultaneously minimizes the distillation loss and CMI of the synthetic dataset in the feature space. This enables the class-aware complexity of synthetic dataset could be efficiently reduced, while distilled data becoming more focused around their class centers.

- Experimental results show that our method can effectively improve the performance of existing dataset distillation methods by up to $5.5\%$. Importantly, our method can be deployed as an plug-and-play module for all the existing DD methods with different optimization objectives.

## 2 RELATED WORKS

Dataset Distillation (DD) (Wang et al., 2018) aims to generate a synthetic dataset that, when used for training, can achieve performance similar to training on the full real dataset. To achieve this, DD adopted a meta-learning process comprising two nested loops. The method minimizes the loss function of the synthetic dataset in the outer loop, using a model trained on the synthetic dataset in the inner loop. To address the issue of unrolled computational graphs, recent studies propose matching proxy information. DC (Zhao et al., 2021) and DCC (Lee et al., 2022) minimize the distance between the gradients of the synthetic and original data on the network being trained with synthetic data. DM (Zhao & Bilen, 2021a) and CAFE Wang et al. (2022) matche the extracted features between the synthetic dataset and the real dataset. MTT (Cazenavette et al., 2022) and TESLA (Cui et al., 2023) match the training trajectories of the network parameters obtained from training on the complete and synthetic datasets, respectively. SRe²L (Yin et al., 2024) successfully scales up dataset distillation to larger datasets and deeper network architectures by utilizing model inversion loss and matching the statistics in batch-normalization layers.

Subsequently, various techniques have been proposed to enhance the performance of the synthetic dataset. DSA (Zhao & Bilen, 2021b) applied differentiable siamese augmentations to both the real and synthetic data while training. IDC (Kim et al., 2022) proposed a multi formulation framework to generate more augmented examples under the same memory budget, successfully achieved a more efficient utilization of the limited pixel space by reducing the resolution of generated images. Since then, incorporating differentiable data augmentation and multi formulation have been adopted by almost all subsequent studies. DREAM (Liu et al., 2023) utilized clustering to select more representative original images. MIM4DD (Shang et al., 2024) proposed leveraging a contrastive learning framework to maximize the mutual information between the synthetic dataset and the real dataset. Several methods have been improved based on the characteristics of the matching objectives. PDD (Chen et al., 2024) and SeqMatch (Du et al., 2024) propose multi-stage distillation based on the characteristics of network parameter variations during the stochastic gradient descent process. IID (Deng et al., 2024) further aligned the inter-class and intra-class relationships when using feature matching.

Existing methods primarily focus on achieving more precise estimation and utilization of surrogate information or compressing additional beneficial information into the synthetic dataset, while neglecting the impact of synthetic dataset complexity. In contrast, we show that properly constraining the complexity of the synthetic dataset yields superior performance.

## 3 NOTATION AND PRELIMINARIES

### 3.1 NOTATION

For a positive integer $K$, we denote $[K] \triangleq \{1, \dots, K\}$. Let $\mathcal{P}([C])$ be the set of all possible probability distributions over the $C$-dimensional probability simplex indexed by $[C]$. For any two probability distributions $P_1, P_2 \in \mathcal{P}([C])$, the cross-entropy between $P_1$ and $P_2$ is defined as $H(P_1, P_2) = -\sum_{i=1}^{C} P_1(i) \log P_2(i)$, and the Kullback–Leibler (KL) divergence between $P_1$ and $P_2$ is defined as $D(P_1 \| P_2) = \sum_{i=1}^{C} P_1(i) \log \frac{P_1(i)}{P_2(i)}$.

For a random variable $X$, let $\mathbb{P}_X$ denote its probability distribution, and $\mathbb{E}_X[\cdot]$ represent the expected value with respect to $X$. For two random variables $X$ and $Y$, let $\mathbb{P}_{(X,Y)}$ denote the joint distribution of $X$ and $Y$. The mutual information between the random variables $X$ and $Y$ is defined as $I(X; Y) = H(X) - H(X \mid Y)$, where $H(X)$ is the entropy of $X$. The conditional mutual information of $X$ and $Z$ given a third random variable $Y$ is defined as $I(X; Z \mid Y) = H(X \mid Y) - H(X \mid Z, Y)$.

Given a deep neural network $f_\theta(\cdot)$ parameterized by $\theta$, we can view it as a mapping from $\mathbf{x} \in \mathbb{R}^d$ to $P_{\mathbf{x}}$. When there is no ambiguity, we denote $P_{\mathbf{x}}$ as $P_{\mathbf{x},\theta}$, and $P_{\mathbf{x}}(y)$ as $P(y \mid \mathbf{x}, \theta)$ for any $y \in [C]$. Let $\hat{Y}$ denote the output predicted by the DNN, with the probability $P_X(\hat{Y})$ in response to the input $X$. Specifically, for any input $\mathbf{x} \in \mathbb{R}^d$, we have $P(\hat{Y} = \hat{y} \mid X = \mathbf{x}) = P_{\mathbf{x}}(\hat{y}) = P(\hat{y} \mid \mathbf{x}, \theta)$. Since $Y \to X \to \hat{Y}$ forms a Markov chain, we can infer that $Y$ and $\hat{Y}$ are independent conditioned on $X$.

## 3.2 DATASET DISTILLATION

Given a large-scale dataset $\mathcal{T} = \{(\mathbf{x}_i, y_i)\}_{i=1}^{|\mathcal{T}|}$, dataset distillation aims to generate a synthetic dataset $\mathcal{S} = \{(\mathbf{s}_i, y_i)\}_{i=1}^{|\mathcal{S}|}$ that retains as much class-relevant information as possible, where $|\mathcal{S}| \ll |\mathcal{T}|$. The key motivation for dataset distillation is to create an informative $\mathcal{S}$ that allows models to achieve performance within an acceptable deviation $\epsilon$ from those trained on $\mathcal{T}$. This can be formulated as:

$$\sup_{(\mathbf{x},y) \in \mathcal{T}} |l(\phi_{\theta_{\mathcal{T}}}(\mathbf{x}), y) - l(\phi_{\theta_{\mathcal{S}}}(\mathbf{x}), y)| \leq \epsilon, \tag{1}$$

where $l(\cdot, \cdot)$ represents the loss function, $\theta_{\mathcal{T}}$ is the parameter of the neural network $\phi$ trained on $\mathcal{T}$, and a similar definition applies to $\theta_{\mathcal{S}}$:

$$\theta_{\mathcal{T}} = \arg\min_{\theta} \mathbb{E}_{(\mathbf{x},y) \in \mathcal{T}} \left( l(\phi_{\theta}(\mathbf{x}), y) \right). \tag{2}$$

To transform this metric into a computable optimization method, previous optimization-based approaches utilize various $\phi(\cdot)$ to extract informative guidance from $\mathcal{T}$ and $\mathcal{S}$ in a specific information space while alternately optimizing $\mathcal{S}$, formulated as:

$$\mathcal{S}^* = \arg\min_{\mathcal{S}} \mathcal{M}(\phi(\mathcal{S}), \phi(\mathcal{T})), \tag{3}$$

where $\mathcal{M}(\cdot, \cdot)$ denotes various matching metrics, such as neural network gradients (Zhao et al., 2021), extracted features (Zhao & Bilen, 2021a), and training trajectories (Cazenavette et al., 2022).

# 4 METHODOLOGY

## 4.1 CLASS-AWARE CMI AS A VALID MEASURE OF SYNTHETIC DATASET

Prior data evaluation studies (Paul et al., 2021; Zheng et al., 2022) indicate that when $|\mathcal{S}|$ is small, the model's ability to learn complex representations is constrained. To enable the model to effectively capture dominant patterns across different classes in a limited dataset, the samples in $\mathcal{S}$ must represent the most prevalent patterns. By focusing on learning from these representative samples, the model can capture essential discriminative information across classes. Additionally, easier samples of certain classes in $\mathcal{S}$ tend to yield lower loss function values, facilitating faster convergence and enhancing performance within a limited number of training iterations. Therefore, it becomes essential to introduce a measure for the class-aware complexity of the synthetic dataset $\mathcal{S}$ itself.

Recognizing that feature representation contains more effective information than probabilistic representation $\hat{Y}$ (Gou et al., 2021), we impose constraints on the synthetic dataset in the feature space. To be specific, for a pre-trained network $f_{\theta^*}(\cdot)$ trained on the original large-scale dataset $\mathcal{T}$, we can decompose it as $f_{\theta^*}(\cdot) = l_{\theta_2^*}(\cdot) \circ h_{\theta_1^*}(\cdot)$ ($\theta^* = \{\theta_1^*, \theta_2^*\}$), where $h_{\theta_1^*}(\cdot) : \mathbf{s} \mapsto \mathbf{z}$ denotes a feature extractor that maps an input sample $\mathbf{s} \in \mathcal{S} \subseteq \mathbb{R}^d$ into an $M$-dimensional feature vector $\mathbf{z} \in \mathcal{Z} \subseteq \mathbb{R}^M$ (e.g., the 512-dimensions penultimate features of ResNet-18), and $l_{\theta_2^*}(\cdot) : \mathbf{z} \mapsto \hat{y}$ is a parametric classifier that takes $\mathbf{z}$ as input and produces a class prediction $\hat{y}$.

For a given input $\mathbf{s} \in \mathcal{S}$, the output feature $\mathbf{z}$ is a deterministic feature vector. We apply the softmax function to the feature vector $\mathbf{z} = (z^1, z^2, \ldots, z^M)$ for an input sample $\mathbf{s} \in \mathcal{S}$, which maps it to a one-dimensional random variable $\hat{Z}$, whose probability distribution $P_{\mathbf{s}}[\hat{Z}]$ is a random point in the probability simplex $\mathcal{P}([M])$ indexed by $[M]$, where

$$P_{\mathbf{s}}[i] = P(\hat{Z} = i \mid \mathbf{s}) = \frac{\exp(z^i)}{\sum_{j=1}^{M} \exp(z^j)}, \quad \text{for any } i \in [M] = \{1, 2, \ldots, M\}. \tag{4}$$

Based on the above analysis, we can see that $Y \to S \to \hat{Z}$ forms a Markov chain. To quantify the class-aware complexity of the synthetic dataset, we introduce the class-aware conditional mutual information (CMI) $I(S; \hat{Z} \mid Y)$ as a novel measure of the complexity of the synthetic dataset and further explain its validity below.

Note that the non-linear relationship between the input $S$ and the output $\hat{Z}$ can be quantified by the conditional mutual information $I(S; \hat{Z} \mid Y)$. This can also be expressed as $I(S; \hat{Z} \mid Y) = H(S \mid$

$Y) - H(S \mid \hat{Z}, Y)$, representing the difference between the uncertainty of $S$ given both $\hat{Z}$ and $Y$ and that of $S$ given $Y$. When a relatively diverse and large dataset (e.g., $\mathcal{T} \sim \mathbb{P}_X$) is used as the input to $f_{\theta^*}(\cdot)$, the corresponding output $\hat{Z}$ follows a more certain probability distribution produced by $f_{\theta^*}(\cdot)$. In contrast, since $S$ is more challenging for randomly initialized networks to learn, its output $\hat{Z}$ often contains excessive confused information related to it, leading to a significant reduction in $H(S \mid \hat{Z}, Y)$. Thus, minimizing the class-aware CMI value constraints the uncertainty brought to $\hat{Z}$ with $S$ as the input of $f_{\theta^*}(\cdot)$, preventing $S$ from overly complicating the prediction process of $f_{\theta^*}(\cdot)$. Additionally, minimizing CMI leads to a reduction in $H(S \mid Y)$ implicitly, which represents the complexity of $S$ conditioned on $Y$ alone.

## 4.2 ESTIMATING THE CLASS-AWARE CMI FOR SYNTHETIC DATASET

Given $Y = y$ and $y \in [C]$, the input $S$ is conditionally distributed according to $P_{S|Y}(\cdot|y)$ and then mapped into $P_S \in \mathcal{P}([M])$. The conditional distribution $P_{\hat{Z}|y}$, i,e., the centroid of this cluster, is exactly the average of input distribution $P_S$ with respect to $P_{S|Y}(\cdot|y)$, which is formulated as:

$$P_{\hat{Z}|y} = \mathbb{E}[P_S \mid Y = y]. \tag{5}$$

Similar to the calculation in (Ye et al., 2024), we employ the Kullback-Leibler (KL) divergence $D(P_S \| P_{\hat{Z}|y})$ to quantify the distance between $P_S$ and the conditional distribution $P_{\hat{Z}|y}$.

Then we can derive the class-aware conditional mutual information as follows:

$$
\begin{aligned}
I(S; \hat{Z} \mid Y = y) &= \sum_{\mathbf{s} \in \mathcal{S}} P_{S|Y}(S = \mathbf{s} \mid y) \left[ \sum_{i=1}^{M} P(\hat{Z} = i \mid \mathbf{s}) \times \ln \frac{P(\hat{Z} = i \mid \mathbf{s})}{P_{\hat{Z}|y}(\hat{Z} = i \mid Y = y)} \right] \quad (6) \\
&= \mathbb{E}_{S|Y} \left[ \left( \sum_{i=1}^{M} P_S[i] \ln \frac{P_S[i]}{P_{\hat{Z}|y}(\hat{Z} = i \mid Y = y)} \right) \middle| Y = y \right] \quad (7) \\
&= \mathbb{E}_{S|Y} \left[ D\left( P_S \| P_{\hat{Z}|y} \right) \mid Y = y \right], \quad (8)
\end{aligned}
$$

Averaging $I(S; \hat{Z}|y)$ with respect to the distribution $P_Y(y)$ of $Y$, we can obtain the conditional mutual information between $S$ and $\hat{Z}$ given $Y$ as follow:

$$\mathrm{CMI}(\mathcal{S}) \triangleq I(S; \hat{Z} \mid Y) = \sum_{y \in [C]} P_Y(y) I(S; \hat{Z} \mid y). \tag{9}$$

In practice, the joint distribution $P(\mathbf{s}, y)$ of $(S, Y)$ may be unknown. To compute the $\mathrm{CMI}(\mathcal{S})$ in this case, we approximate $P(\mathbf{s}, y)$ by the empirical distribution of synthetic dataset $\mathcal{S}_y = \{(\mathbf{s}_1, y), (\mathbf{s}_2, y), \cdots, (\mathbf{s}_n, y)\}$ for any $y \in [C]$. Then the $\mathrm{CMI}_{\mathrm{emp}}(\mathcal{S})$ can be calculated as:

$$\mathrm{CMI}_{\mathrm{emp}}(\mathcal{S}) = \frac{1}{|\mathcal{S}|} \sum_{y \in [C]} \sum_{\mathbf{s}_j \in \mathcal{S}_y} \mathrm{KL}\left( P_{\mathbf{s}_j} \| Q_{\mathrm{emp}}^y \right), \tag{10}$$

$$\text{where} \quad Q_{\mathrm{emp}}^y = \frac{1}{|\mathcal{S}_y|} \sum_{\mathbf{s}_j \in \mathcal{S}_y} P_{\mathbf{s}_j}, \text{ for } y \in [C].$$

## 4.3 DATASET DISTILLATION WITH CMI ENHANCED LOSS

According to the above calculation of CMI, we propose the CMI enhanced Loss $\mathcal{L}$. Overall, it includes two parts:

$$\mathcal{L} = \mathcal{L}_{DD} + \lambda \, \mathrm{CMI}_{\mathrm{emp}}(\mathcal{S}). \tag{11}$$

The first term $\mathcal{L}_{DD}$ represents any loss function in previous DD methods, e.g., DM, DSA and MTT , $\lambda > 0$ is a weighting hyperparameter.

The proposed CMI enhanced loss is universal since it provides a plug-and-play solution to minimize the class-conditional complexity of the synthetic dataset, making it adaptable to all previous dataset distillation methods that focus on better aligning the synthetic dataset with the real one just based on

Table 1: Comparative analysis of dataset distillation methods. Δ: the improvement magnitude of CMI as a plugin to the base distillation methods. Ratio (%): the proportion of condensed images relative to the number of entire training set. Whole Dataset: the accuracy of training on the entire original dataset. The **best** results are highlighted.

| Method | SVHN | | CIFAR10 | | CIFAR100 | |
|---|---|---|---|---|---|---|
| IPC | 10 | 50 | 10 | 50 | 10 | 50 |
| Ratio (%) | 0.14 | 0.7 | 0.2 | 1 | 2 | 10 |
| Random | 35.1±4.1 | 70.9±0.9 | 26.0±1.2 | 43.4±1.0 | 14.6±0.5 | 30.0±0.4 |
| Herding(Welling, 2009) | 50.5±3.3 | 72.6±0.8 | 31.6±0.7 | 40.4±0.6 | 17.3±0.3 | 33.7±0.5 |
| K-Center(Sener & Savarese, 2017) | 14.0±1.3 | 20.1±1.4 | 14.7±0.9 | 27.0±1.4 | 17.3±0.2 | 30.5±0.3 |
| Forgetting(Toneva et al., 2019) | 16.8±1.2 | 27.2±1.5 | 23.3±1.0 | 23.3±1.1 | 15.1±0.2 | 30.5±0.4 |
| MTT (Cazenavette et al., 2022) | 79.9±0.1 | 87.7±0.3 | 65.3±0.4 | 71.6±0.2 | 39.7±0.4 | 47.7±0.2 |
| MIM4DD (Shang et al., 2024) | - | - | 66.4±0.2 | 71.4±0.3 | 41.5±0.2 | - |
| SeqMatch (Du et al., 2024) | 80.2±0.6 | 88.5±0.2 | 66.2±0.6 | **74.4±0.5** | 41.9±0.5 | **51.2±0.3** |
| **MTT+CMI** | **80.8±0.2** | **88.8±0.1** | **66.7±0.3** | 72.4±0.3 | 41.9±0.4 | 48.8±0.2 |
| Δ | (0.9↑) | (1.1↑) | (1.4↑) | (0.8↑) | (2.2↑) | (1.1↑) |
| DM (Zhao & Bilen, 2023) | 72.8±0.3 | 82.6±0.5 | 48.9±0.6 | 63.0±0.4 | 29.7±0.3 | 43.6±0.4 |
| IID-DM (Deng et al., 2024) | 75.7±0.3 | **85.3±0.2** | **55.1±0.1** | 65.1±0.2 | 32.2±0.5 | 43.6±0.3 |
| **DM+CMI** | **77.9±0.4** | 84.9±0.4 | 52.9±0.3 | **65.8±0.3** | **32.5±0.4** | **44.9±0.2** |
| Δ | (5.1↑) | (2.3↑) | (4.0↑) | (2.8↑) | (2.8↑) | (1.3↑) |
| IDM (Zhao et al., 2023) | 81.0±0.1 | 84.1±0.1 | 58.6±0.1 | 67.5±0.1 | 45.1±0.1 | 50.0±0.2 |
| IID-IDM (Deng et al., 2024) | 82.1±0.3 | 85.1±0.5 | 59.9±0.2 | 69.0±0.3 | 45.7±0.4 | 51.3±0.4 |
| **IDM+CMI** | **84.3±0.2** | **88.9±0.2** | **62.2±0.3** | **71.3±0.2** | **47.2±0.4** | **51.9±0.3** |
| Δ | (3.3↑) | (4.8↑) | (3.6↑) | (3.8↑) | (2.1↑) | (1.9↑) |
| DSA (Zhao et al., 2021) | 79.2±0.5 | 84.4±0.4 | 52.1±0.5 | 60.6±0.5 | 32.3±0.3 | 42.8±0.4 |
| **DSA+CMI** | **80.5±0.2** | **85.5±0.3** | **54.7±0.4** | **66.1±0.1** | **35.0±0.4** | **45.9±0.3** |
| Δ | (1.3↑) | (1.1↑) | (2.6↑) | (5.5↑) | (2.7↑) | (3.1↑) |
| IDC (Kim et al., 2022) | 87.5±0.3 | 90.1±0.1 | 67.5±0.5 | 74.5±0.1 | 45.1±0.4 | - |
| DREAM (Liu et al., 2023) | 87.9±0.4 | 90.5±0.1 | 69.4±0.4 | 74.8±0.1 | **46.8±0.7** | 52.6±0.4 |
| PDD (Chen et al., 2024) | - | - | 67.9±0.2 | 76.5±0.4 | 45.8±0.5 | 53.1±0.4 |
| **IDC+CMI** | **88.5±0.2** | **92.2±0.1** | **70.0±0.3** | **76.6±0.2** | 46.6±0.3 | **53.8±0.2** |
| Δ | (1.0↑) | (2.1↑) | (2.5↑) | (2.1↑) | (1.5↑) | - |
| Whole Dataset | 95.4±0.2 | | 84.8±0.1 | | 56.2±0.3 | |

feature similarity. In addition, minimizing CMI as a complexity constraint poses certain challenges, as the optimization target is a general synthetic dataset. To address this issue, we compute CMI in the feature space rather than the probability space as demonstrated in (Ye et al., 2024), thereby establishing a stronger constraint. Additionally, during the optimization process, we randomly sample pre-trained networks with varying initializations to compute CMI, resulting in a dataset-specific optimization process that is independent of the network parameters.

## 5 EXPERIMENTS

In this section, we conduct comprehensive experiments to evaluate our proposed method across various datasets and network architectures. We begin by detailing the implementation, followed by an analysis of the performance improvements achieved with different distillation techniques. Finally, we assess the effectiveness of the the CMI enhanced loss through a series of ablation studies.

### 5.1 EXPERIMENTAL SETTINGS

**Datasets.** We conduct our experiments on five standard datasets with varying scales and resolutions: SVHN(Sermanet et al., 2012) CIFAR-10/100 (Krizhevsky, 2009), Tiny-ImageNet (Le & Yang, 2015) and ImageNet-1K(Deng et al., 2009). SVHN contains over $600,000$ images of house numbers from around the world. CIFAR-10 and CIFAR-100 consist of $50,000$ training images, with 10 and 100 classes, respectively. The image size for CIFAR is $32 \times 32$. For larger datasets, Tiny-ImageNet contains $100,000$ training images from 200 categories, with the image size of $64 \times 64$. ImageNet-1K consists of over $1,200,000$ training images with various resolutions from $1,000$ classes.

**Baselines.** We consider state-of-the-art (SOTA) plug-and-play techniques for enhancing dataset distillation performance as baselines, each with different optimization objectives:

- MIM4DD (Shang et al., 2024) introduces a contrastive learning framework to estimate and maximize the mutual information between the synthetic dataset and the real dataset.
- SeqMatch (Du et al., 2024) implements dynamic adjustment of the optimization process by splitting the synthetic dataset in gradient matching methods.
- IID (Deng et al., 2024) better aligns the intra-class and inter-class relationships between the original dataset and the synthetic dataset in feature matching methods.
- DREAM (Liu et al., 2023) universally accelerates dataset distillation by selecting representative original images through clustering.
- PDD (Chen et al., 2024) generates a considerably larger synthetic dataset by aligning the parameter variations throughout the gradient matching process.

**Architectures.** Our experimental settings follow that of Cazenavette et al. (2022) and SRe$^2$L Yin et al. (2024): we employ a ConvNet for distillation, with three convolutional blocks for CIFAR-10 and CIFAR-100 and four convolutional blocks for Tiny-ImageNet respectively, each containing a 128-kernel convolutional layer, an instance normalization layer (Ulyanov et al., 2016), a ReLU activation function (Nair & Hinton, 2010) and an average pooling layer. For larger datasets (i.e., ImageNet-1K), we employ ResNet18 (He et al., 2016) as the backbone of SRe$^2$L. To compute CMI, we employ multiple pre-trained ResNet18 trained on each corresponding dataset. AlexNet, VGG11 and ResNet18 are used to assess the cross-architecture performance.

**Evaluation.** Following previous works, we generate synthetic datasets with IPC=1, 10, 50, and 100, then train multiple randomly initialized networks to evaluate their performance. All the training configurations follow the corresponding base distillation methods. We compute the Top-1 accuracy on the test set of the real dataset, repeating each experiment five times to calculate the mean.

Table 2: Synthetic dataset performance(%) under IPC=1. CMI can only be applied to methods that utilize multi-formulation introduced by IDC.

| Method | SVHN | CIFAR10 | CIFAR100 |
|---|---|---|---|
| IDM Zhao et al. (2023) | 65.3±0.3 | 45.6±0.7 | 20.1±0.3 |
| IID-IDM Deng et al. (2024) | 66.3±0.1 | 47.1±0.1 | 24.6±0.1 |
| **IDM+CMI** | **66.4±0.3** | **47.5±0.2** | **24.7±0.5** |
| Δ | (1.1↑) | (1.9↑) | (4.6↑) |
| IDC Kim et al. (2022) | 68.5±0.9 | 50.6±0.4 | - |
| DREAM Liu et al. (2023) | 69.8±0.8 | 51.1±0.3 | 29.5±0.3 |
| **IDC+CMI** | **70.0±0.5** | **51.6±0.3** | **30.1±0.1** |
| Δ | (1.5↑) | (1.0↑) | - |

Table 3: Performance of the CMI enhanced loss on higher resolution datasets. †: the reported error range is reproduced by us.

| Method | TinyImageNet | | |
|---|---|---|---|
| IPC | 1 | 10 | 50 |
| MTT Cazenavette et al. (2022) | 8.8±0.3 | 23.2±0.2 | 28.0±0.3 |
| **MTT+CMI** | - | **24.1±0.3** | **28.8±0.3** |
| Δ | - | (1.9↑) | (0.8↑) |
| IDM Zhao et al. (2023) | 9.8±0.2 | 21.9±0.2 | 26.2±0.3 |
| **IDM+CMI** | **10.4±0.3** | **23.7±0.3** | **27.7±0.1** |
| Δ | (0.6↑) | (1.8↑) | (1.5↑) |
| IDC Kim et al. (2022) | 9.5±0.3† | 24.5±0.4† | 29.0±0.2† |
| **IDC+CMI** | **10.4±0.3** | **25.7±0.3** | **30.1±0.1** |
| Δ | (0.9↑) | (1.2↑) | (1.1↑) |

## 5.2 PERFORMANCE IMPROVEMENT

We demonstrate the effectiveness of our proposed method by applying the CMI-enhanced loss to previous optimization-based distillation methods on CIFAR-10/100, Tiny-ImageNet, and ImageNet-1K. Our approach is compared with state-of-the-art baselines for various distillation methods on low-resolution datasets, with IPC set to 10 and 50 for all benchmarks, as using only one distilled image per class makes empirical computation of CMI infeasible. Additionally, we evaluate methods that employ multi-formulation techniques, such as IDM. As shown in Table 1 and Table 2, our method consistently yields significant performance improvements across different datasets and compression ratios, with modest gains in trajectory matching and substantial gains in feature and gradient matching methods. Notably, CMI + DSA surpasses DSA by $5.5\%$ on CIFAR-10 with IPC=50.

For higher-resolution datasets, we conducted experiments on representative optimization-based distillation methods with various objectives on Tiny-ImageNet. As shown in Table 3, noticeable improvements were achieved under different settings. For ImageNet-1K, we applied the CMI enhanced loss to SRe$^2$L, which decouples the distillation process by using a pre-trained model and optimizes the synthetic

Table 4: Performance improvement with SRe$^2$L on ImageNet-1K.

| Method | ImageNet-1K | | |
|---|---|---|---|
| IPC | 10 | 50 | 100 |
| SRe$^2$L Yin et al. (2024) | 21.3±0.6 | 46.8±0.2 | 52.8±0.3 |
| **SRe$^2$L+CMI** | **24.2±0.3** | **49.1±0.1** | **54.6±0.2** |
| Δ | (2.9↑) | (2.3↑) | (1.8↑) |

Table 5: Cross-architecture performance on CIFAR10 under IPC=10 with different distillation methods.

| Method | AlexNet | VGG11 | ResNet18 |
|---|---|---|---|
| MTT Zhao et al. (2023) | 34.2±2.6 | 50.3±0.8 | 46.4±0.6 |
| **MTT+CMI** | **34.9±1.1** | **51.3±0.6** | **47.7±0.7** |
| Δ | (0.7↑) | (1.0↑) | (1.3↑) |
| IDM Zhao et al. (2023) | 44.6±0.8 | 47.8±1.1 | 44.6±0.4 |
| **IDM+CMI** | **54.0±0.4** | **57.0±0.2** | **53.7±0.2** |
| Δ | (9.4↑) | (9.2↑) | (9.1↑) |
| IDC Kim et al. (2022) | 63.5±0.4† | 64.4±0.4† | 61.2±0.2† |
| **IDC+CMI** | **64.4±0.3** | **67.4±0.2** | **66.0±0.4** |
| Δ | (0.9↑) | (3.0↑) | (4.8↑) |

Table 6: Performance obtained simultaneously applying previous techniques and CMI enhance loss on different distillation methods.

| Method | SVHN | CIFAR10 | CIFAR100 |
|---|---|---|---|
| MTT Cazenavette et al. (2022) | 79.9±0.1 | 65.3±0.4 | 39.7±0.4 |
| SeqMatch Du et al. (2024) | 80.2±0.6 | 66.2±0.6 | 41.9±0.5 |
| **SeqMatch+CMI** | **81.2±0.4** | **67.3±0.5** | **43.1±0.3** |
| IDM Zhao et al. (2023) | 81.0±0.1 | 58.6±0.1 | 45.1±0.1 |
| IID Deng et al. (2024) | 82.1±0.3 | 59.9±0.2 | 45.7±0.4 |
| **IID+CMI** | **85.1±0.3** | **63.0±0.3** | **47.9±0.3** |
| IDC Kim et al. (2022) | 87.5±0.3 | 67.5±0.5 | 45.1±0.4 |
| DREAM Liu et al. (2023) | 87.9±0.4 | 69.4±0.4 | 46.8±0.7 |
| **DREAM+CMI** | **88.2±0.1** | **71.5±0.1** | **47.9±0.3** |

dataset through cross-entropy loss and batch normalization statistics. This enables it to handle large datasets and achieve reliable performance. However, SRe$^2$L faces the same challenge of excessive dataset complexity as other optimization-based distillation methods. Following the original settings, we validate the effectiveness of our proposed method with the results shown in Table 4 for larger synthetic datasets (e.g., IPC=100).

### 5.3 CROSS-ARCHITECTURE GENERALIZATION

We evaluated the performance of architectures different from the backbone used to distill CIFAR-10. Following the settings in previous works, our experiments included widely used models such as AlexNet, VGG11, and ResNet18, with consistent optimizers and hyperparameters. Following the same pipeline used to evaluate the same-architecture performance, each experiment is repeated five times and the mean values are reported as shown in Table 5.

We can see that the considerable cross-architecture generalization performance improvement achieved by our method, especially when using IDM and IDC as the basic distillation method. By leveraging pre-trained models with diverse architectures in the distillation phase, reducing the CMI value effectively constrains the complexity of the synthetic dataset. The results indicate that our synthetic dataset is robust to changes in network architectures.

### 5.4 COMBINATION WITH DIFFERENT TECHNIQUES

Unlike existing dataset distillation techniques, our method achieves stable performance improvements by reducing the complexity of the synthetic dataset itself. Furthermore, Our approach can serve as a unified technique added to existing methods, and experiments have demonstrated its generality. Table 6 demonstrates the effectiveness of combining existing techniques with CMI enhanced loss on different basic distillation methods on CIFAR-10 with IPC=10. It can be observed that different optimization directions enable our method to be integrated with other techniques, leading to considerable performance improvements.

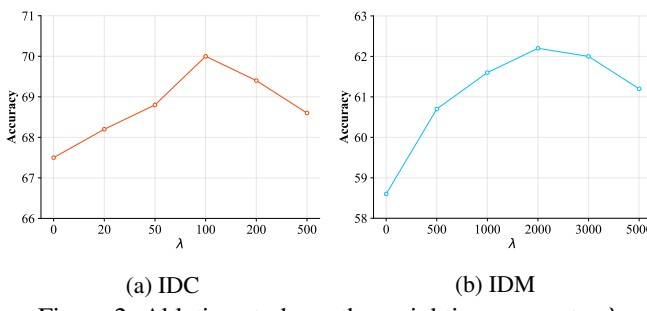

(a) IDC      (b) IDM

Figure 2: Ablation study on the weighting parameter λ.

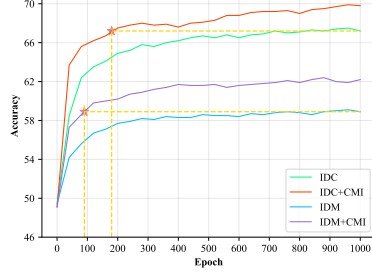

Figure 3: Accuracy curve *w.* and *w.o.* CMI constraint.

### 5.5 ABLATION STUDIES AND ANALYSIS

We conducted additional experiments to verify the effectiveness of CMI enhanced loss. Unless otherwise specified, the experiments were conducted on CIFAR-10 with IPC=10.

**Sample distribution.** As defined in Equation (5), the centroid of the DNN's outputs from a specific class $Y = y$, where $y \in [C]$, is the average of $P_X$ with respect to the conditional distribution $\mathbb{P}_{(X|Y)}(\cdot | Y = y)$. Referring to Section 4.2, fixing the label $Y = y$, the $\mathrm{CMI}(f, Y = y)$ measures the distance between the centroid $Q^y$ and the output $P_X$ for class $y$. Thus, $\mathrm{CMI}(f, Y = y)$ indicates how concentrated the output distributions $P_X$ are around the centroid $Q^y$ for class $y$.

To more intuitively demonstrate how the strategy of minimizing CMI values affects the complexity of the synthetic dataset, we visualize the t-SNE graphs of the synthetic dataset in both feature space and probability space using a pre-trained ConvNet. As shown in Figure 1, even when evaluating the dataset using different network architectures, the clusters for the synthetic dataset with lower CMI values are more concentrated. Furthermore, reducing CMI values in the feature space also effectively concentrates the dataset clusters around the class center in probability space.

**Evaluation of hyper-parameter $\lambda$.** We conducted a sensitivity analysis focusing on the weighting parameter $\lambda$ in Section 4.3. The results shown in Figure 2 indicate that the CMI value is too small compared to $\mathcal{L}_{DD}$ in practical applications, requiring a larger $\lambda$ to achieve the desired constraint effect. To achieve this, the value of $\lambda$ must be adjusted according to $\mathcal{L}_{DD}$. However, a too large $\lambda$ often shifts the optimization objective towards clustering the synthetic dataset around the centroids, leading to degraded performance. Varying $\lambda$ between 500 and 5000 resulted in only a 1.5% performance difference on IDM, indicating moderate sensitivity.

**Training Efficiency.** In addition to the stable performance improvements shown in Table 1, we visualized the accuracy curve during training by applying the CMI enhanced loss with different distillation methods. As shown in Figure 3, unlike previous methods that typically require extensive training iterations to converge, our proposed method achieves comparable performance with significantly fewer iterations (e.g., one-fifth and one-tenth of the epochs required on IDC and IDM respectively) and further improve the performance with additional iterations, more visualizations can be seen in Appendix A.3.

**CMI Constraint Performance under different Settings.** We explore multiple settings for computing the empirical CMI value to more comprehensively assess the effectiveness of CMI enhanced loss: (1) *Pre-trained*: we investigate whether it is necessary to pre-train the surrogate model for computing CMI value on the corresponding dataset; (2) *Architecture*: we compute CMI using surrogate models with different network architectures; (3) *Space*: whether computing CMI in probability space or feature space leads to better constraint effects.

Table 7: Ablation study on different settings of computing CMI value. PT indicates pre-trained or not.

| Space | PT | Arch ConvNet | ResNet18 |
|---|---|---|---|
| Probability | - | 67.7±0.1 | 67.0±0.4 |
| | ✓ | **68.1±0.2** | **67.8±0.3** |
| Feature | - | 67.3±0.2 | 67.9±0.4 |
| | ✓ | **68.7±0.3** | **70.0±0.3** |

Table 7 illustrates the results of computing and minimizing the CMI value under different settings. We observe that selecting an effective surrogate network architecture for CMI calculation and pre-training the model is essential, as it allows the model to generate more accurate class centroids for computing CMI values. Additionally, better results are achieved by computing CMI in feature space rather than in probability space. We hypothesize that computing CMI in probability space makes the constraint more dependent on the certain model architecture. Note that to align with the baseline, all settings related to pre-training are consistent with IID (Deng et al., 2024).

## 5.6 VISUALIZATIONS

We compare the distillation results with and without the proposed CMI enhanced loss in Figure 4. Unlike the t-SNE results in Figure 1, the constraint has minimal impact on the generated images, with differences nearly imperceptible to the human eye. This intriguing phenomenon suggests that CMI enhances performance without making substantial changes to the pixel space, offering a method that differs from existing approaches by having minimal impact on the images. Additional visualizations comparison of the distilled datasets with different distillation methods are provided in Appendix A.8.

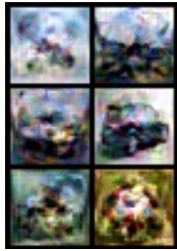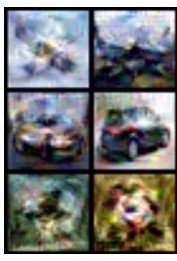

Figure 4: Visualization comparison between DSA (Left column) and DSA with CMI constraint (Right column).

# 6 CONCLUSION

In this paper, we present a novel conditional mutual information (CMI) enhanced loss for dataset distillation by analyzing and reducing the class-aware complexity of synthetic datasets. The proposed method computes and minimizes the empirical conditional mutual information of a pre-trained model, effectively addressing the challenges faced by previous dataset distillation approaches. Extensive experiments demonstrate the effectiveness of the CMI constraint in enhancing the performance of existing dataset distillation methods across various benchmark datasets. This novel approach emphasizes the intrinsic properties of the dataset itself, contributing to an advanced methodology for dataset distillation.

**Acknowledgement.** This work is supported in part by the National Natural Science Foundation of China under grant 62171248, 62301189, Peng Cheng Laboratory (PCL2023A08), and Shenzhen Science and Technology Program under Grant KJZD20240903103702004, JCYJ20220818101012025, RCBS20221008093124061, GXWD20220811172936001.

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

# A  APPENDIX

## A.1  COMPARISON WITH RELATED WORKS

Several recent works also focus on optimizing the dataset distillation process from the perspective of the synthetic dataset itself. IID (Deng et al., 2024) proposes using the L2 norm to aggregate the features of the synthetic dataset, thereby constraining its classification boundary. SDC (Wang et al., 2024a) suggests reducing the complexity of the synthetic dataset using

Table 8: Performance comparison with SDC under different settings.

| Dataset | SVHN | | TinyImageNet | |
|---|---|---|---|---|
| Method | DSA | | MTT | |
| IPC | 10 | 50 | 10 | 50 |
| SDC (Wang et al., 2024a) | 79.4±0.4 | 85.3±0.4 | 20.7±0.2 | 28.0±0.2 |
| +CMI | **80.5±0.2** | **85.5±0.3** | **24.1±0.3** | **28.8±0.3** |

GraDN-Score (Paul et al., 2021) constraints. However, IID restricts its method to distribution matching, and the L2 norm constraints fail to provide stable improvements due to the problem of dimensionality explosion. Notably, the performance comparison in Table 1 is based on IID, which additionally optimizes the variance matching between the synthetic and original datasets.

Although SDC shares a similar starting point with our approach, it merely uses the gradient of the distillation loss as a constraint and restricts its method to gradient matching. This approach is constrained by the proxy backbone, leading to weaker constraint effects and limited performance improvements due to the same optimization objective with the distillation loss. A comparison of experimental results is shown in Table 8.

## A.2  REGULARIZATION PROPRIETY

Here, we analyze the trends of dataset distillation loss $\mathcal{L}_{DD}$ during training as shown in Section 4.3. $\mathcal{L}_{DD}$ refers to any dataset distillation loss. By incorporating CMI as a regularization term during dataset distillation, we successfully constrained the complexity of the synthetic dataset. As shown in Figure 5, we present the $\mathcal{L}_{DD}$ curves throughout the training process. It can be observed that adding CMI as a constraint decreased the main function losses, demonstrating that our proposed method effectively reduces the complexity of the synthetic dataset.

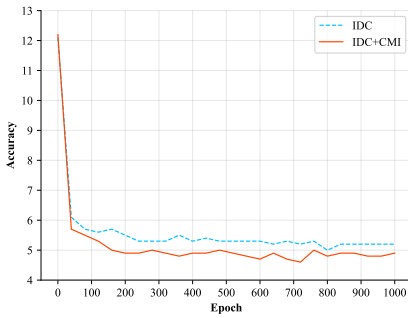

Figure 5: The $\mathcal{L}_{IDC}$ curves while training on CIFAR10 under IPC=10.

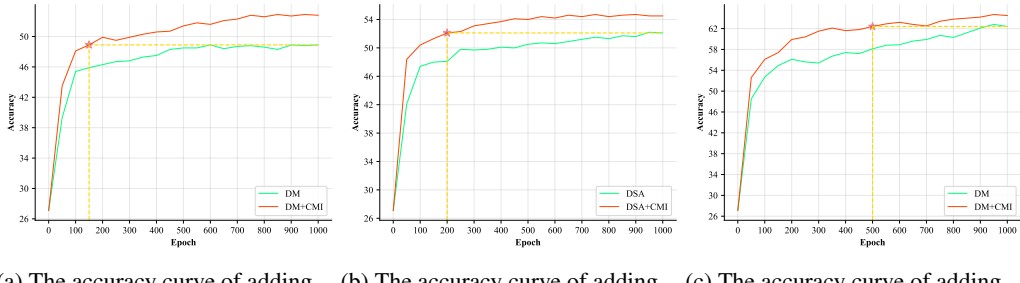

(a) The accuracy curve of adding CMI constraint to DM.

(b) The accuracy curve of adding CMI constraint to DSA.

(c) The accuracy curve of adding CMI constraint to MTT.

Figure 6: Applying CMI constraint brings stable performance and efficiency improvements with different optimization objectives.

## A.3  EFFICIENT TRAINING PROCESS

By deploying the CMI constraint, we not only achieve performance improvements but also significant computational acceleration. As shown in Figure 6, we present the accuracy curves after incorporating the CMI constraint on DSA (Zhao & Bilen, 2021b), DM (Zhao & Bilen, 2021a), and MTT (Cazenavette et al., 2022). It can be seen that our method achieves considerable computational acceleration with the same initialization settings. These results indicate that actively constraining the

complexity of the synthetic dataset during training benefits both new network training and reduces gradient bias, leading to faster convergence.

---

**Algorithm 1** Disentangled computation of applying CMI constraint

---

**Input**: $\mathcal{L}$: distillation loss; $f_{\theta^*}$: pre-trained classifier; $\mathcal{T}$: real dataset; $\phi(\cdot)$: proxy information extractor; $\mathcal{M}$: distance metric; $K$: training iteration number $C$: class number;

1: $\mathcal{L} = 0$
2: Randomly initialize $\mathcal{S}$
3: **for** i $\leftarrow$ 0 to $K - 1$ **do**
4:     Randomly sample $f_{\theta^*}$
5:     $\mathcal{L} = \mathcal{L} + \mathcal{M}(\phi(\mathcal{S}), \phi(\mathcal{T}))$
6:     **for** j $\leftarrow$ 0 to $C - 1$ **do**
7:         $\text{CMI}_{\text{emp\_c}}(\mathcal{S}) = \sum_{\mathbf{s} \in \mathcal{S}_j} \text{KL}\left(P_{\mathbf{s}} \| Q_{\text{emp}}^j\right)$
8:         $\mathcal{L} = \mathcal{L} + \lambda \cdot \text{CMI}_{\text{emp\_c}}(\mathcal{S})$
9:     **end for**
10:    $\mathcal{S} \leftarrow SGD(\mathcal{S}; \mathcal{L})$
11: **end for**

**Output**: Synthetic dataset $\mathcal{S}$

---

### A.4 DISENTANGLED CMI VALUE COMPUTATION

Simultaneously computing and minimizing the CMI value for the entire synthetic dataset during distillation can lead to significant memory consumption as IPC and the number of categories increase. To address this issue, we propose decomposing the CMI computation shown in Section 4.2 by category and embedding it into the original distillation process, as shown in Algorithm 1. Since the computation of the empirical CMI value is category-independent, the decoupled calculation can equivalently represent the original equation without any loss of fidelity. This approach greatly alleviates the memory overhead while achieving consistent results.

### A.5 ADDITIONAL TRAINING COST

We further analyze the extra time consumption caused by applying CMI constraint in Table 9 on a single NVIDIA GeForce RTX 3090. For IDM, adding the CMI constraint nearly doubles the time consumption; however, considering that we need only one-tenth to one-quarter of the distillation iterations to achieve the original performance, using the CMI constraint can save up to 50% of the time. In contrast, for IDC, the additional time cost introduced by the CMI constraint is negligible compared to the method

Table 9: Additional time consumption (s) per iteration of adding CMI constraint.

| Method | CIFAR10 | | CIFAR100 | |
|---|---|---|---|---|
| IPC | 10 | 50 | 10 | 50 |
| IDM (Zhao et al., 2023) | 0.5 | 0.6 | 5.3 | 7.1 |
| IDM+CMI | 1.0 | 1.2 | 10.2 | 11.2 |
| IDC (Kim et al., 2022) | 18.2 | 22.4 | 190.5 | 215.7 |
| IDC+CMI | 22.6 | 24.2 | 222.5 | 236.8 |

itself. Moreover, since applying CMI enhanced loss can reduce the number of distillation iterations to one-fifth of the iterations to achieve the original performance, deploying CMI is justified. Notably, we calculate and optimize CMI at each iteration. However, to save computational resources, we could reduce CMI calculations to once every five iterations while maintaining performance, thus speeding up distillation.

### A.6 ABLATION STUDY ON MODEL ARCHITECTURES

To further investigate the impact of teacher model architecture, we conduct more comprehensive ablation experiments. Under the CIFAR10 with IPC=10, we utilized simply pre-trained ConvNet, AlexNet, ResNet18, and VGG11 as proxy models to compute CMI for both DM and DSA.

We sequentially report the following metrics: the depth (i.e., number of convolution layer), the dimensionality $M$ of the feature space, and the accuracy of the simply pre-trained proxy model; the start CMI value of the synthetic dataset, the CMI value at the end of the optimization process w.o. and

Table 10: Ablation study of model architecture on DM.

| Method | ConvNet | AlexNet | VGG11 | ResNet18 |
|---|---|---|---|---|
| Depth | 3 | 5 | 8 | 17 |
| Dimension | 2048 | 4096 | 512 | 512 |
| Model Acc (%) | 79.8 | 80.2 | 79.4 | 79.5 |
| Start value | 0.1045 | 1.9338 | 0.0021 | 0.0016 |
| End value w.o. CMI | 0.0866 | 1.6217 | 0.0059 | 0.0066 |
| End value w. CMI | 0.0326 | 0.6642 | 0.0006 | 0.0004 |
| Performance (%) | 51.2 | 50.8 | 52.4 | **52.9** |

Table 11: Ablation study of model architecture on DSA.

| Method | ConvNet | AlexNet | VGG11 | ResNet18 |
|---|---|---|---|---|
| Depth | 3 | 5 | 8 | 17 |
| Dimension | 2048 | 4096 | 512 | 512 |
| Model Acc (%) | 79.8 | 80.2 | 79.4 | 79.5 |
| Start value | 0.1045 | 1.9338 | 0.0021 | 0.0016 |
| End value w.o. CMI | 0.0825 | 1.8755 | 0.0076 | 0.0051 |
| End value w. CMI | 0.0455 | 0.8875 | 0.0005 | 0.0004 |
| Performance (%) | 53.2 | 53.4 | **54.8** | 54.7 |

w. CMI constraint of the synthetic dataset, and the accuracy of the model trained using the synthetic dataset. As shown in table below, it can be observed that the dimensionality of the feature space has a greater impact on CMI values than network depth, and CMI effectively reflects the degree of clustering in $\hat{Z}$ under the same dimensional conditions. Nonetheless, the proposed CMI constraint effectively regulates the complexity of the synthetic dataset, demonstrating the strong generalization capability of our method.

## A.7 LIMITATION AND FUTURE WORK

**IPC=1.** Minimizing CMI is equal to reduce the Kullback-Leibler (KL) divergence $D(\cdot\|\cdot)$ between $P_S$ and $P_{\hat{Z}|y}$, hence, when the IPC is 1, the CMI value is 0.

**Proxy Model.** Computing CMI requires choosing an appropriate proxy model architecture to provide $\hat{Z}$ when $S$ is used as the input, and selecting an architecture often relies on heuristic methods. Besides, similar to IID (Deng et al., 2024), constraining the synthetic datasets with CMI needs multiple pre-trained teacher networks, which leads to a substantial consumption overhead.

Here, we suggest a future direction to better utilize the proxy model; mixing the proxy model could potentially provide a more informative CMI constraint. And One can try to perturb a single model to obtain multiple teacher models which is similar to what DWA (Du et al., 2025) did.

Recent years, some dataset distillation methods have introduced GANs as a parameterization technique (Cazenavette et al., 2023; Zhong et al., 2024a). Moreover, with the successful application of diffusion models in various fields (Zhu et al., 2024b;a), some dataset distillation methods have begun using diffusion models to directly generate images (Gu et al., 2024a; Su et al., 2024; Zhong et al., 2024b). In future work, we will explore the application of CMI to these types of dataset distillation methods.

## A.8 DISTILLED DATASET VISUALIZATION

To demonstrate the effectiveness of our method, we present the t-SNE visualization of the synthetic dataset with the CMI constraint on DSA, as shown in Figure 7. It can be observed that the CMI constraint provides strong regularization effects in both feature space and probability space, resulting in clearer class boundaries. Additionally, we visualize the effects of synthetic datasets produced using different datasets and distillation methods. As shown in Figures 8 to 11, the CMI constraint not only significantly enhances performance but also introduces minimal perceptual distortion.

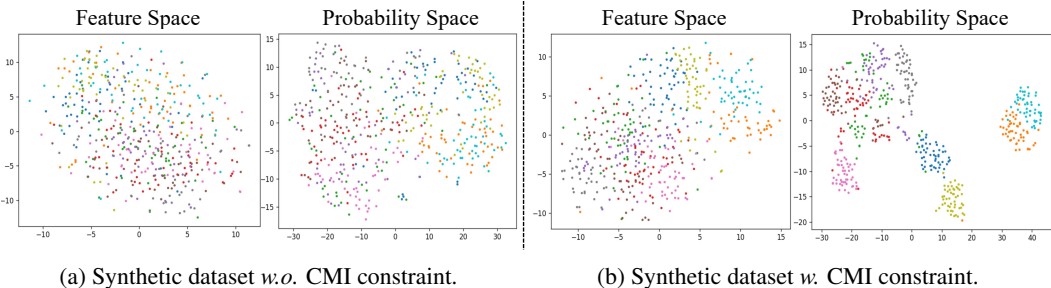

(a) Synthetic dataset *w.o.* CMI constraint.   (b) Synthetic dataset *w.* CMI constraint.

Figure 7: Visualization of the synthetic dataset generated by DSA with (a) high CMI value, and (b) low CMI value.

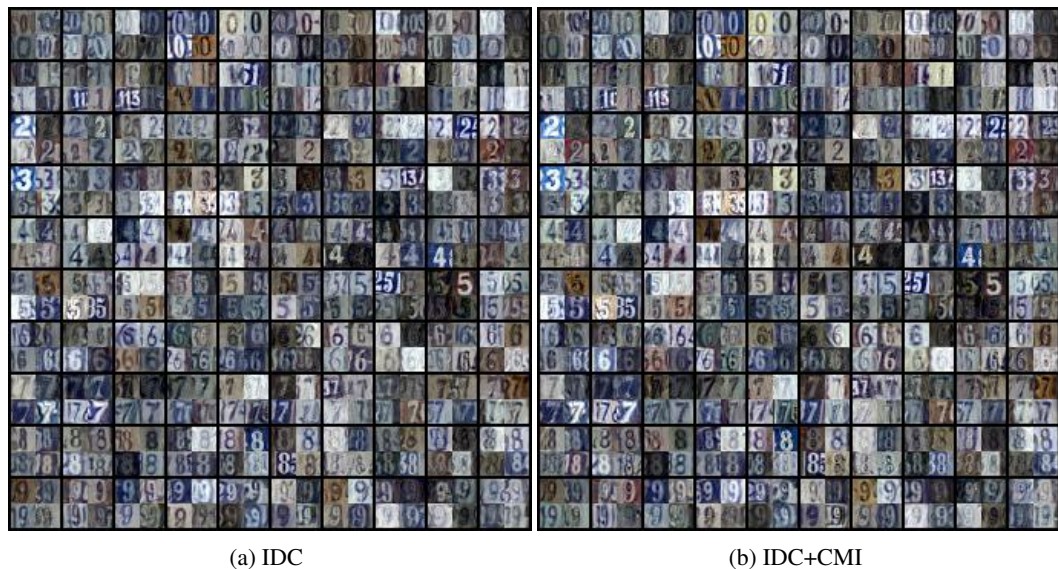

(a) IDC                                           (b) IDC+CMI

Figure 8: Visualization of the synthetic dataset generated by IDC on SVHN.

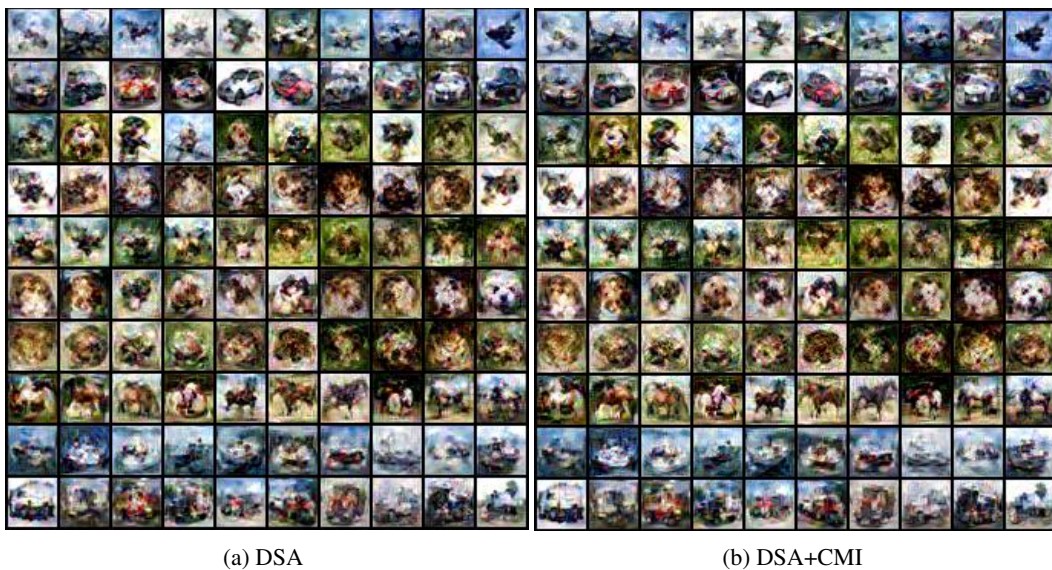

(a) DSA                                           (b) DSA+CMI

Figure 9: Visualization of the synthetic dataset generated by DSA on CIFAR10.

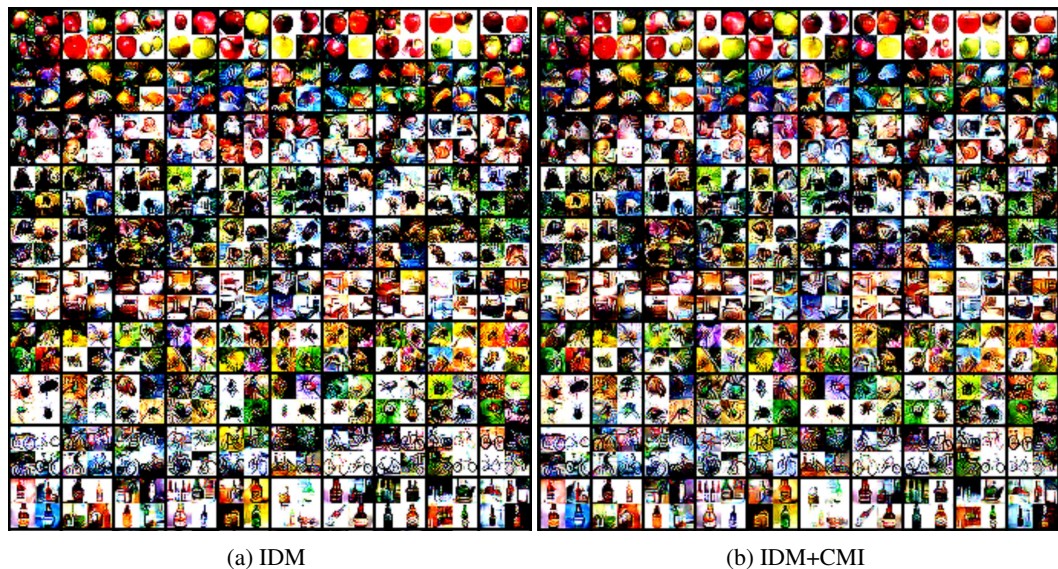

(a) IDM                                            (b) IDM+CMI

Figure 10: Partial visualization of the synthetic dataset generated by IDM on CIFAR100.

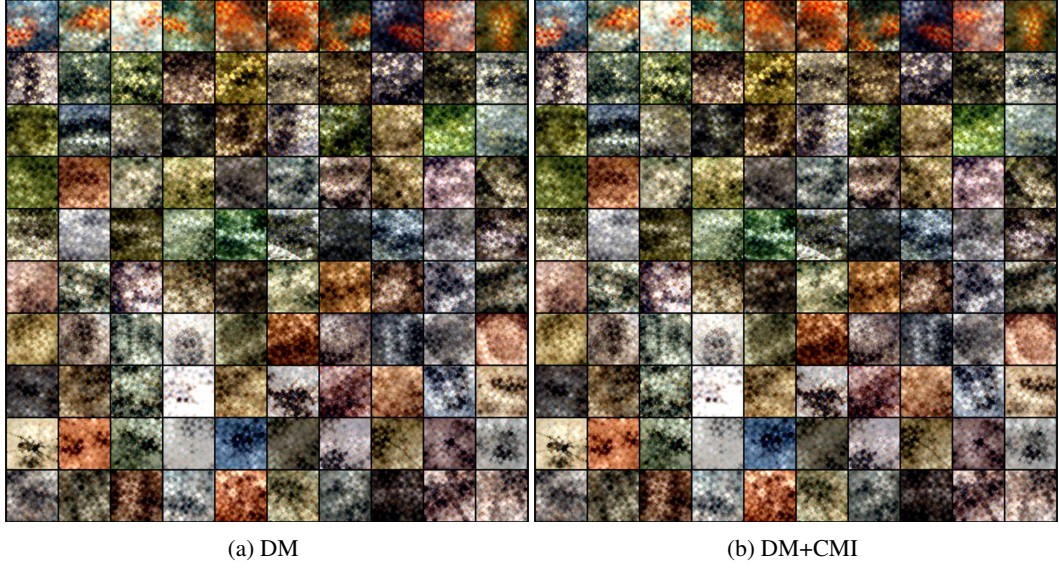

(a) DM                                            (b) DM+CMI

Figure 11: Partial visualization of the synthetic dataset generated by DM on TinyImageNet.

