# OpenReview forum: "Going Beyond Feature Similarity: Effective Dataset distillation based on Class-aware Conditional Mutual Information"
_ICLR.cc/2025/Conference — ICLR 2025 Poster_

### Official Review · Reviewer_Zh5i · 2024-11-02

**Soundness:** 2
**Presentation:** 3
**Contribution:** 2
**Rating:** 3
**Confidence:** 4

**Summary:**

This paper proposes a plug-and-play method, termed CMI, designed to enhance existing DD techniques by minimizing conditional mutual information. By applying CMI, the distilled data is concentrated more effectively around the center of each class, thereby improving generalizability.

**Strengths:**

- The logic is clear.
- The experiments are comprehensive.
- The review of related works is thorough.
- The proposed method is theoretically sound.

**Weaknesses:**

- The core ideas, methodology, and formulations in this paper draw substantially from the approach proposed in [1].
- If  \hat{Z}  contains excessive confusing or uninformative information related to  S , then  H(S | \hat{Z}, Y)  will not be reduced; rather, it could remain the same or even increase. This is because conditional entropy reflects the remaining uncertainty in  S  after observing both  \hat{Z}  and  Y . When  \hat{Z}  is noisy or irrelevant for predicting  S , it does not help in reducing this uncertainty.
- Line 213 states that “minimizing the class-aware CMI reduces the uncertainty brought to  \hat{Z}  conditioned on  S ,” which should be  “minimizing the class-aware CMI reduces the uncertainty brought to S conditioned on  \hat{Z}”.
- The authors’ derivation of Equation 6 lacks an explicit explanation, making it challenging to fully understand the transition from previous formulations.
- Works like [2] and [3], which also target improvements in dataset distillation, are not adequately considered.
- Equation 3 requires summing over all synthetic instances within class  y , how the authors adapt this approach to instance-based distillation methods like SRe2L.

[1] Bayes conditional distribution estimation for knowledge distillation based on conditional mutual information

[2]TOWARDS LOSSLESS DATASET DISTILLATION VIA DIFFICULTY-ALIGNED TRAJECTORY MATCHING

[3]Prioritize Alignment in Dataset Distillation

**Questions:**

see weakness.

---

> ### Author Response · Authors · 2024-11-21
>
> We would like to express our gratitude for your valuable suggestions and insightful feedback of our paper. Our point-by-point responses are provided below.
>
> >**Weakness 1: Contribution of our work.**
>
> Thank you for acknowledging the previous work [1] that utilizes conditional mutual information (CMI). Here, we present the distinctions between our proposed method and [1] as follows:
>
> **Different Optimization Objectives.** [1] lies in the field of knowledge distillation, **with the goal of enabling the teacher model to be a better estimated Bayesian classifier**. While our proposed method is applied to evaluate the non-linear information between $\mathcal{S}$ and $\hat{Z}$ and to **make the synthetic dataset a more learnable dataset**.
>
> **Different Optimization Intentions.** [1] **maximize** the CMI value to help the teacher model better capture the contextual information during **pre-training phase**. In contrast, our proposed method seeks to **minimize** the CMI value of the synthetic dataset to constrain its complexity during **distillation process**.
>
> **Different Optimization Methods.** [1] optimizes by calculating CMI in the **probability space**, whereas our approach reconstructs the Markov chain (i.e., from $Y \rightarrow X \rightarrow \hat{Y}$ to $Y \rightarrow S \rightarrow \hat{Z}$) in **feature space** to compute CMI and validate the theoretical correctness as discussed in **Sec. 4.1**. A stronger constraint effect is achieved as shown in **Table 7** provided in **Sec 5.5**.
>
> Beyond that, we emphasize that the derivation process of CMI stems from a **deterministic** calculation in information theory, that is to say specific calculation is deterministic since the variable is determined. This is why our formulation is similar to [1] (Despite significant differences discussed above). Thus, the focus when applying CMI as a regularization should be why and how to determine the three corresponding random variables.
>
> On the other hand, similar to the way of our work does, a series of studies employing methodologies utilizing information theory to enhance performance across various tasks have been widely applied, including in dataset distillation [2] [3] [4]. In contrast to the various estimation framework necessitated by the **nondeterministic** calculation of other information metric (e.g., mutual information), we reclaim that the major advantage of CMI is its certainty, i.e., the only estimation derived from the distribution sampling.
>
> In summary, **our study is the first to identify the intrinsic class-aware complexity challenge in dataset distillation and to introduce the CMI as a solution**, elucidating its theoretical and practical significance. Furthermore, We offer a novel perspective on how information theory can be effectively applied to this field, thereby highlighting the originality and advancement of our research.
>
> [1] Bayes conditional distribution estimation for knowledge distillation based on conditional mutual information. In ICLR, 2024.
>
> [2] MIM4DD: Mutual Information Maximization for Dataset Distillation. In NeurIPS, 2023.
>
> [3] One Training Fits All: Generalized Data Condensation via Mixture-of-Information Bottleneck Guidance. OpenReview Submission, 2024.
>
> [4] GIFT: Unlocking Full Potential of Labels in Distilled Dataset at Near-zero Cost. Openreview Submission, 2024.

---

> ### Author Response · Authors · 2024-11-21
>
> >**Weakness 2: Confusion about the computation of conditional entropy $H(S|\hat{Z}, Y)$**
>
> We truly appreciate your detailed concerns about CMI for enhancing the rigor of the paper. The objective of minimizing the CMI is to mainly reduce the difference between conditional entropy $H(S \mid Y)$ and $H(S \mid \hat{Z}, Y)$,  rather than simply minimizing $H(S \mid \hat{Z}, Y)$. Moreover, introducing additional conditions (i.e., the random variable $\hat{Z}$) will either keep the entropy constant or decrease it, and it cannot lead to an increase as stated. This is a fundamental fact of information theory.
>
> In fact, $\hat{Z}$ is a function of the random variable $S$, where $S$ serves as the input of the deterministic $f_{\theta^*}(\cdot)$ and $\hat{Z}$ is the output. At this point, if the goal is to minimize $H(S \mid \hat{Z}, Y)$, $f_{\theta^*}$ should not change any input, leading to $H(S \mid \hat{Z}, Y) = H(S \mid S, Y) = 0$.
>
> On the other hand, our research lies in the field of dataset distillation, where the optimization objective does not lead to the synthetic dataset becoming entirely noise, which has also been discussed in recent work [5]. To certify this assumption, we used different pre-trained classifier to evaluate the synthetic datasets distilled by different distillation methods as shown in Table below. For a more comprehensive understanding, we also present the performance of CMI constraint using different proxy models in Table.
>
> | Method        | ConvNet  |  | AlexNet  |  | ResNet18  |  | VGG11  |  |
> |---------------|---------------|---------------|---------------|---------------|----------------|----------------|------------|------------|
> |          | 10         | 50         | 10          | 50         |  10           | 50          | 10      | 50      |
> | MTT           | 98.0          | 100.0         | 97.0          | 99.0          | 98.0           | 100.0          | 100.0      | 100.0      |
> | MTT + CMI     | 100.0         | 100.0         | 100.0         | 100.0         | 100.0          | 100.0          | 100.0      | 100.0      |
> | DSA           | 72.0          | 82.6          | 79.0          | 83.8          | 71.0           | 72.2           | 75.0       | 78.4       |
> | DSA + CMI     | 100.0         | 100.0         | 99.0          | 100.0         | 100.0          | 100.0          | 100.0      | 100.0      |
> | DM            | 47.0          | 87.6          | 59.0          | 85.8          | 56.0           | 79.6           | 60.0       | 78.8       |
> | DM + CMI      | 99.0          | 100.0         | 98.0          | 100.0         | 100.0          | 100.0          | 100.0      | 100.0      |
>
> >**Weakness 3: Ambiguous Expression**
>
> Thank you for pointing out the issue of unclear expression in this section. Here, the point we aim to convey is ''Minimizing the class-aware CMI value constraints the uncertainty brought to $\hat{Z}$ with $\mathcal{S}$ as the input of $f_{\theta^{*}}(\cdot)$'', instead of $H(\hat{Z} \mid \mathcal{S})$. The statement will be revised in the final version of the manuscript to ensure greater precision, and we sincerely appreciate your feedback in pointing this out.
>
> >**Weakness 4: Lack of explicit explanation of CMI computation**
>
> Thanks for your suggestion, here, we provide a more detailed mathematical derivation. We first define the input synthetic data $\mathcal{S}$ is conditionally distributed according to $P_{S \mid Y}(\cdot \mid y)$ and mapped into $P_{S} \in \mathcal{P}([M])$. Then we can formulate $P_{\hat{Z} \mid Y}$ as the average of $P_{S}$ with respect to $P_{S \mid Y}(\cdot \mid y)$, which is shown in **Eq. 5**:
> $$
> P_{\hat{Z} \mid y } = \mathbb{E}[ P_S \mid Y =y ].
> $$
>
> We get the conditional mutual information $I(S ; \hat{Z} \mid Y=y)$ between $S$ and $\hat{Z}$ given $Y$:
>
> $$
> I(S ; \hat{Z} \mid Y) = \sum_{y \in[C]} P_{Y}(y) I(S ; \hat{Z} \mid y) = \mathbb{E}\left[D\left(P_S \| P_{\hat{Z} \mid Y}\right) \right].
> $$
>
> Given $Y = y$, the Kullback-Leibler (KL) divergence $D(\cdot \| \cdot)$ between $P_{S}$ and $P_{\hat{Z} \mid y}$ is equal to **Eq. 6**:
>
> $$
> \mathbb{E}\left[D\left(P_{S} \| P_{\hat{Z} \mid y}\right) \mid Y=y\right] = \mathbb{E}\left[\left(\sum_{i=1}^{M} P_{S}(i) \ln \frac{P_{S}(i)}{P_{\hat{Z} \mid y}(\hat{Z}=i \mid Y=y)}\right)\mid Y=y\right]
> $$
>
> $$
> = \sum_{S} P_{S \mid Y}(\mathbf{s} \mid y)\left[\sum_{i=1}^{M} P(\hat{Z}=i \mid \mathbf{s}) \ln \frac{P(\hat{Z}=i \mid \mathbf{s})}{P_{\hat{Z} \mid y}(\hat{Z}=i \mid Y=y)}\right].
> $$
>
> These details will be incorporated into the final version of the manuscript to enhance its rigor, and we sincerely appreciate your constructive feedback.
>
> [5] What is Dataset Distillation Learning? In ICLR, 2024.

---

> ### Author Response · Authors · 2024-11-21
>
> >**Weakness 5: Comparison with recent SOTA.**
>
> Thanks for your suggestion. In contrast to [6] and [7], which are exclusively designed for trajectory matching, (i.e., selecting specific range of trajectories), both our proposed method and the majority baseline methods function as a versatile plugin (e.g., DREAM [8] serves as a universal solution applicable to various distillation methods). Thus, we consider applying the CMI constraint to [6] as suggested by `R#kma8`. As demonstrated in table below, our proposed approach consistently delivers substantial performance enhancements.
>
> | Method        | CIFAR10 |  | CIFAR100  |  | TinyImageNet  |  |
> |---------------|---------------|---------------|----------------|----------------|--------------------|--------------------|
> |        | 10    | 50    | 10     | 50    | 10         | 50         |
> | MTT           | 65.3 ± 0.4    | 71.6 ± 0.2    | 39.7 ± 0.4     | 47.7 ± 0.2     | 23.2 ± 0.2         | 28.0 ± 0.3         |
> | **MTT+CMI**   | **66.7 ± 0.3**| **72.4 ± 0.3**| **41.9 ± 0.4** | **48.8 ± 0.2** | **24.1 ± 0.3**     | **28.8 ± 0.3**     |
> | DATM          | 66.8 ± 0.2    | 76.1 ± 0.3    | 47.2 ± 0.4     | 54.1 ± 0.2     | 31.1 ± 0.3         | 39.7 ± 0.3         |
> | **DATM+CMI**  | **67.4 ± 0.4**| **76.6 ± 0.1**| **47.6 ± 0.3** | **55.1 ± 0.3** | **31.9 ± 0.4**     | **40.6 ± 0.2**     |
>
> >**Weakness 6: Details about the implementation of SRe$^2$L.**
>
> Thank you for pointing out the confused expression. **Eq. 3** in our paper represents an abstraction of optimization-based dataset distillation methods, wherein current methods indirectly optimize **Eq. 2** by aligning specific information (e.g., feature distribution). As discussed in [9] and [10]. instance-based methods (i.e., SRe$^2$L) still falls under this category.
>
> SRe$^2$L optimizes the synthetic dataset by minimizing the loss function $L = l(\phi_{\theta_{\mathcal{T}}}(\mathbf{s}), y)$, where $y \in [C]$ and $l$ represents the cross entropy loss. Due to the instance-independent characteristic of its optimization process, we transit the optimization from the inner loop to the outer loop, enabling simultaneous optimization of the entire synthetic dataset. During the optimization process, synthetic dataset is classified based on predefined labels which are used to compute the loss function, enabling the deployment of CMI constraint.
>
> The change of pseudocode is shown below. For a fair comparison, we present the impact of altering the loop order in table below. We attribute the performance improvement to the more precise class boundaries achieved by applying CMI constraint for datasets generated by SRe$^2$L, a critical aspect that has also been emphasized in recent studies [10] [11] [12] [13], and our method can be more easily applied on recent SOTA [12] and [13] with their class-wise supervision.
>
> | Method                     | 10        | 50        | 100       |
> |----------------------------|-----------|-----------|-----------|
> | SRe$^2$L  | 21.3 ± 0.6| 46.8 ± 0.2| 52.8 ± 0.3|
> | SRe$^2$L$^\dagger$ | 21.5 ± 0.3| 46.3 ± 0.4| 53.4 ± 0.1|
> | **SRe$^2$L + CMI** | **24.2 ± 0.3**| **49.1 ± 0.1**| **54.6 ± 0.2**|
>
> ### Original Pseudocode
>
> ```markdown
> for i from 1 to IPC:  # Outer loop optimizes synthetic dataset from 1 to IPC
>     targets = np.arange(1000) # initial hard label
>     for j from 1 to T:  # Inner loop optimizes images from 1 to epoch T
>         Optimize S[i:, ] with L
> ```
> ### Modified Pseudocode
> ```markdown
> targets = np.tile(np.arange(1000), (ipc, 1) )# initial hard label
> for i from 1 to T:  # Outer loop optimizes synthetic dataset from 1 to epoch T
>     for j from 1 to IPC:  # Inner loop optimizes images from 1 to IPC
>         Optimize S[j, :] with L
>     for k from 1 to C:  # apply CMI constraint for each class
>         Optimize S[:, k] with CMI constraint
> ```
>
> [6] Towards Lossless Dataset Distillation via Difficulty-Aligned Trajectory Matching. In ICLR, 2024.
>
> [7] Prioritize Alignment in Dataset Distillation. In arXiv, 2024.
>
> [8] Dream: Efficient Dataset Distillation by Representative Matching. In ICCV, 2023.
>
> [9] Generalized Large-Scale Data Condensation via Various Backbone and Statistical Matching. In CVPR, 2024.
>
> [10] Elucidating the Design Space of Dataset Condensation. In NeurIPS, 2024.
>
> [11] Breaking Class Barriers: Efficient Dataset Distillation via Inter-Class Feature Compensator. Openreview Submission, 2024.
>
> [12] Dataset Distillation via the Wasserstein Metric. In arXiv, 2023.
>
> [13] Are Large-scale Soft Labels Necessary for Large-scale Dataset Distillation? In NeurIPS, 2024.

---

> ### Author Response · Authors · 2024-11-24
>
> Dear Reviewer Zh5i
>
> Thank you once again for dedicating your valuable time to reviewing our paper and providing constructive comments! Considering the limited time available and to save the reviewer's time, we summarize our responses here.
>
> - **1. Contribution of our work:**
>   - We present a clear distinction between our method and existing approaches utilizing CMI.
>   - We highlight that one major advantage of CMI stems from its computational determinism.
>   - we elaborate on the originality of our method in applying information-theoretic principles and outlined its contributions.
>
> - **2. Confusion about the computation of conditional entropy $H(S \mid \hat{Z}, Y)$:**
>   - We point out that incorporating additional observations will only decrease or maintain entropy.
>   - We argue that $\hat{Z}$ would not be irrelevant information in the field of dataset distillation.
>
> - **3. Ambiguous Expression:**
>   - We clarify the ambiguous statements present in the paper.
>
> - **4. Lack of explicit explanation of CMI computation:**
>   - We re-derive the mathematical computation process from the definition of CMI to its computable expansion.
>
> - **5. Comparison with recent SOTA:**
>   - We provide additional experimental results by applying CMI on DATM across various datasets.
>
> - **6. Details about the implementation of SRe$^2$L:**
>   - We demonstrate how to losslessly modify SRe$^2$L code to implement CMI constraints.
>
> As the end of the discussion period approaches, we kindly ask if our responses have satisfactorily addressed your concerns. Your feedback would be greatly appreciated, and we would be delighted to engage in further discussions if needed.
>
> Sincerely,
>
> The Authors

---

> > ### Comment · Reviewer_Zh5i · 2024-11-24
> >
> > Thank you for the explaination.
> >
> > The difference you highlighted pertains to the distinction between KD and DD, rather than the core differentiator between the proposed methods. Both the methods utilize class-aware CMI for optimization, and the only modification in Eq. 7 is the substitution of $X \to S$ and $\hat{Y} \to \hat{Z}$.
> >
> > Furthermore, based on the provided pseudocode, the proposed CMI introduces an additional class-wise optimization step for the generated images, which incurs extra computational cost. Considering the very marginal performance gain, this additional complexity significantly diminishes the overall contribution of the proposed method, especially when compared to other advancements on SRe2L.
> >
> > So I will maintain my score.

---

> ### Author Response · Authors · 2024-11-25
>
> We thank the reviewer for the valuable feedback and provide further clarifications below：
>
> >**1. Utilization of CMI:**
>
> Conditional Mutual Information (CMI), derived from information theory, serves as a information measure of the relationship between two random variables and is **not domain-specific**. Unlike Mutual Information (MI), one significant advantage of CMI lies in its computational determinism. **We believe that appropriately leveraging CMI according to task-specific characteristics across different fields has significantly facilitated the mutual verification of theory and experimentation**, which has also been acknowledged and accepted by `R#oMQb` and `R#kma8`.
>
> >**2. Confusion between KD and DD:**
>
> Here, we **reemphasize** that the differences between knowledge distillation (KD) and dataset distillation (DD) can lead to **substantial** impacts. In contrast to previous work [1], our method introduces clear differences (e.g., optimization direction). We notice that in the feedback, you seem to consider that two random variables  $\hat{Y}$ and $\hat{Z}$ can be straightforwardly substituted when reconstructing Markov chains, while we have made a detailed discussion in **Sec 4.1**. And we have made a clear clarification that the focus when applying CMI as a regularization should be **why and how to determine the three corresponding random variables** and **appropriate optimization intention**.
>
> >**3. Application to SRe$^2$L:**
>
> Our work aims to constrain the excessive complexity in existing dataset distillation tasks by utilizing CMI as a **general plug-and-play regularization** which has been acknowledged by all the reviewers including `R#Zh5i`. Simply conflating our method with methods solely aimed at improving accuracy on SRe$^2$L could be inappropriate. Based on your suggestion, we conduct comparisons between our method and **PAD [2] recommended by the reviewer** on SRe$^2$L. Results on CIFAR-100 are presented in the table below.
>
> | Method                     | 10        | 100       |
> |----------------------------|-----------|-----------|
> | SRe$^2$L  | 28.2| 57.2|
> | SRe$^2$L + PAD (FIEX) | 29.3| 57.9|
> | **SRe$^2$L + CMI** | **30.1**| **59.1**|
>
> >**More effective modification to SRe$^2$L Code:**
>
> We appreciate your valuable advice and have made improvements to our method based on your suggestion and insight from existing SOTA method [3]. By simply swapping class loop with IPC loop, we successfully eliminate additional computational overhead. The revised pseudocode is provided below, and additional experimental results are shown in the table below. Under class-wise supervision **without incurring extra computational cost**, the performance improvements achieved with our method go beyond what could be described as **"very marginal performance gain"**, even compared with recent SOTA plug-and-play method [4] solely designed for instance-based distillation method, further validating the generalization and robustness of our proposed method.
>
> | Method                     | 10        | 50        | 100       |
> |----------------------------|-----------|-----------|-----------|
> | SRe$^2$L  | 21.3 ± 0.6| 46.8 ± 0.2| 52.8 ± 0.3|
> | SRe$^2$L$^\dagger$ | 22.7 ± 0.1| 48.4 ± 0.2| 54.3 ± 0.3|
> | **SRe$^2$L + CMI** | **24.1 ± 0.3 (2.8$\uparrow$)**| **50.3 ± 0.4(3.5$\uparrow$)**| **56.0 ± 0.3(3.2$\uparrow$)**|
>
> ### Original Pseudocode
>
> ```markdown
> for i from 1 to IPC:  # Outer loop optimizes synthetic dataset from 1 to IPC
>     targets = np.arange(C) # initial hard label for all class
>     for j from 1 to T:  # Inner loop optimizes images from 1 to epoch T
>         Optimize S[i:, ] with L
> ```
> ### Modified Pseudocode
> ```markdown
> for i from 1 to C:  # Outer loop optimizes synthetic dataset from 1 to class C
>     targets = np.arange(IPC) # initial hard label for one class
>     for j from 1 to T:  # Inner loop optimizes images from 1 to epoch T
>         Optimize S[i, :] with L and CMI constraint
> ```
>
>
> [1] Bayes conditional distribution estimation for knowledge distillation based on conditional mutual information
>
> [2] Prioritize Alignment in Dataset Distillation. Openreview Submission, 2024.
>
> [3] Are Large-scale Soft Labels Necessary for Large-scale Dataset Distillation? In NeurIPS, 2024.
>
> [4] GIFT: Unlocking Full Potential of Labels in Distilled Dataset at Near-zero Cost. Openreview Submission, 2024.

---

> > ### Author Response · Authors · 2024-11-28
> >
> > Dear Reviewer,
> >
> > In response to your valuable feedback, we have provided clearer definitions, conducted additional experiments, and presented new results. If you still have any questions, we are eager to hear details. We encourage you to tell us any questions you still have, and we promise to address them in detail. Thank you.
> >
> > Sincerely,
> >
> > The Authors

---

> ### Author Response · Authors · 2024-12-01
>
> Dear Reviewer Zh5i
>
> We appreciate your response and the contribution your feedback has made to improving our work! As the end of the discussion period approaches, if all your queries have been addressed, we kindly ask you to consider raising your rating. If you still have any doubts or reservations about our work, we are more than willing to engage in further discussion with you.
>
> Sincerely,
>
> The Authors

---

### Official Review · Reviewer_i6mf · 2024-11-03

**Soundness:** 3
**Presentation:** 2
**Contribution:** 3
**Rating:** 6
**Confidence:** 3

**Summary:**

This paper introduces a novel regularization method for dataset distillation (DD) by minimizing both the distillation loss and Conditional Mutual Information (CMI) of synthetic datasets. It uses an efficient CMI estimation method to measure class-aware properties and combines CMI with existing DD techniques. Experiments show that the proposed CMI-enhanced loss significantly outperforms state-of-the-art methods, improving performance by up to 5.5%. The method can be used as a plug-and-play module for all existing DD methods with diverse optimization objectives.

**Strengths:**

The strengths of this paper lie in its comprehensive experimentation across diverse datasets and network architectures, which effectively demonstrates the versatility and robustness of the proposed method. Furthermore, the method's ability to be integrated as a plug-and-play module into existing dataset distillation techniques, regardless of their optimization objectives, showcases its innovation and flexibility, making it a significant contribution to the field.

**Weaknesses:**

The paper lacks a clear discussion of the limitations of the proposed method. Furthermore, the authors should consider using more intuitive explanations, visual aids, and pseudocode to help readers better understand the technical details of the method.

**Questions:**

Can you discuss any potential limitations of your proposed method and suggest directions for future work to address these limitations?

---

> ### Author Response · Authors · 2024-11-21
>
> Thank you for dedicating your time and effort to reviewing our paper. We are deeply encouraged by your positive comments and recognition of our work. Below, we provide detailed responses to address your concerns.
>
> >**Question 1: Limitation and future work.**
>
> Thank you for your positive comments and valuable suggestions. Here, we outline some limitations of our proposed method, we will add the corresponding issues in the final version of the manuscript and include them in the future work section.
>
> **Proxy Model.** Computing CMI requires an proxy model to provide $\hat{Z}$ when $\mathcal{S}$ is used as the input, and selecting an architecture often relies on heuristic methods.
>
> **IPC=1.** Minimizing CMI is equal to reduce the Kullback-Leibler (KL) divergence $ D(\cdot \|\| \cdot) $ between $ P_{S} $ and  $ P_{\hat{Z} \mid y} $, hence, when the IPC is 1, the CMI value is 0.
>
> Here, we suggest a future direction to better utilize the proxy model; mixing the proxy model discussed in [1] could potentially provide a more informative CMI constraint. While for the IPC=1 setting, we suggest using multi-formation introduced in [2] and [3], or other parameterization techniques. Corresponding experimental results are shown in **Table 2** provided in **Sec 5.2**.
>
> [1] Four eyes see more than two: Dataset Distillation with Mixture-of-Experts. OpenReview Submission, 2024.
>
> [2] Dataset Condensation via Efficient Synthetic-Data Parameterization. In ICML, 2022.
>
> [3] Improved Distribution Matching for Dataset Condensation. In CVPR, 2023.

---

> ### Author Response · Authors · 2024-11-24
>
> Dear Reviewer i6mf
>
> Thank you once again for dedicating your valuable time to reviewing our paper and providing constructive comments! Considering the limited time available and to save the reviewer's time, we summarize our responses here.
>
> - **1. Limitation and future work:**
>   - We identify the limitations in the current deployment of CMI constraint.
>   - We propose the direction of future work to address corresponding issues.
>
> As the end of the discussion period approaches, we kindly ask if our responses have satisfactorily addressed your concerns. Your feedback would be greatly appreciated, and we would be delighted to engage in further discussions if needed.
>
> Sincerely,
>
> The Authors

---

> ### Author Response · Authors · 2024-12-01
>
> Dear Reviewer i6mf
>
> We appreciate your response and the contribution your feedback has made to improving our work! As the end of the discussion period approaches, if all your queries have been addressed, we kindly ask you to consider raising your rating. If you still have any doubts or reservations about our work, we are more than willing to engage in further discussion with you.
>
> Sincerely,
>
> The Authors

---

### Official Review · Reviewer_kma8 · 2024-11-04

**Soundness:** 3
**Presentation:** 3
**Contribution:** 3
**Rating:** 6
**Confidence:** 4

**Summary:**

This paper proposes a Conditional Mutual Information (CMI) method as a plug-and-play loss function to enhance the performance of dataset distillation methods. Experiments conducted on multiple baseline methods demonstrate the effectiveness of the CMI loss.

**Strengths:**

1.	The proposed CMI method is a relatively simple yet effective approach that is plug-and-play in nature. It has demonstrated its effectiveness across multiple baseline methods.
2.	The motivation behind the method proposed in the paper is solid and is supported by a certain theoretical foundation.
3.	The experiments in the paper are comprehensive, conducted across various scales of datasets.

**Weaknesses:**

1.	There are now newer and more powerful methods available, such as "Towards Lossless Dataset Distillation via Difficulty-Aligned Trajectory Matching" (ICLR 2024). The authors could consider experimenting with their proposed method on these methods.
2.	The description of the method in the paper could be clearer, particularly regarding the explanation of the formula symbols, to better emphasize the key points of the approach. Currently, it appears somewhat ambiguous.
3.	In my view, using mutual information or KL divergence is not a particularly novel approach, as it has been employed in many works across various fields.

**Questions:**

Please kindly refer to the above weaknesses.

---

> ### Author Response · Authors · 2024-11-21
>
> We sincerely thank you for the precious time and effort in providing a wealth of suggestions to enhance the quality of our paper. We have carefully read all the comments and provide detailed point-by-point responses as follows. Hopefully, we can adequately address your concerns.
> > **Question 1: Additional experiment on more powerful distillation method.**
>
> Thanks for the valuable suggestions on improving and completing the experimental aspects. We conduct an additional experiment by applying the CMI constraint on [1]. The experimental results show that our proposed method, as a general-purpose plugin, can achieve performance improvements across various distillation methods.
>
> | Method        | CIFAR10 |  | CIFAR100  |  | TinyImageNet  |  |
> |---------------|---------------|---------------|----------------|----------------|--------------------|--------------------|
> |        | 10    | 50    | 10     | 50    | 10         | 50         |
> | MTT           | 65.3 ± 0.4    | 71.6 ± 0.2    | 39.7 ± 0.4     | 47.7 ± 0.2     | 23.2 ± 0.2         | 28.0 ± 0.3         |
> | **MTT+CMI**   | **66.7 ± 0.3**| **72.4 ± 0.3**| **41.9 ± 0.4** | **48.8 ± 0.2** | **24.1 ± 0.3**     | **28.8 ± 0.3**     |
> | DATM          | 66.8 ± 0.2    | 76.1 ± 0.3    | 47.2 ± 0.4     | 54.1 ± 0.2     | 31.1 ± 0.3         | 39.7 ± 0.3         |
> | **DATM+CMI**  | **67.4 ± 0.4**| **76.6 ± 0.1**| **47.6 ± 0.3** | **55.1 ± 0.3** | **31.9 ± 0.4**     | **40.6 ± 0.2**     |
>
> >**Question 2: Emphasis on the key point of the methodology.**
>
> Thank you for your suggestions on improving the presentation. Here, we make a briefly summary of our work as follows:
>
> On the basis of observing excessive inter-class complexity in the synthetic dataset, i.e., more entangled $\hat{Z}$ when $\mathcal{S}$ is used as the input of a pre-trained classifier $f_{\theta^{*}}$. we introduce conditional mutual information (CMI) from information theory as a metric to evaluate the complexity of the synthetic dataset. By reconstructing the original CMI computation formula in the feature space and incorporating it as a plug-and-play regularization term, which is subsequently minimized, we achieve stable performance improvements across various distillation methods and datasets. A detailed notation is provided in table below, and we will refine our statements in the final version of the manuscript to ensure greater rigor.
> | Notation              | Description                                             |
> |-----------------------|---------------------------------------------------------|
> | $\mathcal{T}$        | the original training dataset                           |
> | $\mathcal{S}$        | the synthetic training dataset                          |
> | $S$                   | the random variable as the input to the network        |
> | $Y$                   | the random variable as the label                        |
> | $y$                   | some certain label of the random variable $Y$          |
> | $\hat{Z}$             | the random variable as the penultimate features of the network |
> | $C$                   | the number of the class                                    |
> | $M$                   | the number of the dimensionality of $\hat{Z}$         |
> | $f_{\theta^{*}}$     | the pre-trained network                                |
> | $Q_{\text{emp}}^{y}$ | the empirical center of class $y$                      |
> | $\lambda$             | the hyper-parameter of CMI constraint                  |
>
> >**Question 3: Essential difference with other Information theoretic metrics.**
>
> Here, we demonstrate the advantages of CMI over mutual information (MI). In theory, the forward propagation of neural networks $f_{\theta^{*}}$ is a deterministic process, causing $I(X; \hat{Z})$ to degrade into $H(\hat{Z})$, thereby losing its theoretical significance. On the other hand, the value spaces of $X$ and $\hat{Z}$ are extremely large, making $I(X;\hat{Z})$ very difficult to compute and approximate. As a result, variational methods are commonly employed for estimation, but these estimates are often inaccurate in practice. While CMI enables the computation process to follow a deterministic mathematical expansion by incorporating an additional variable $Y$. This results in a more accurate information metric without the need for additional estimation frameworks (e.g., the contrastive learning framework employed in [2]) and provides a universal regularization approach. The quantitative comparisons with [2] are shown in **Table 1** provided in **Sec 5.5**.
>
> [1] Towards Lossless Dataset Distillation via Difficulty-Aligned Trajectory Matching. In ICLR, 2024.
>
> [2] MIM4DD: Mutual Information Maximization for Dataset Distillation. In NeurIPS, 2023.

---

> ### Author Response · Authors · 2024-11-24
>
> Dear Reviewer kma8
>
> Thank you once again for dedicating your valuable time to reviewing our paper and providing constructive comments! Considering the limited time available and to save the reviewer's time, we summarize our responses here.
>
> - **1. Additional experiment on more powerful distillation method:**
>   - We provide additional experimental results by applying CMI on DATM across various datasets.
>
> - **2. Emphasis on the key point of the methodology:**
>   - We concisely summarize the insights and workflow of our proposed method.
>   - We provide annotations for the symbols referenced in the paper.
>
> - **3. Essential difference with other Information theoretic metrics:**
>   - We demonstrate the superiority of CMI over MI based on the principle of the Information Bottleneck theory.
>
> As the end of the discussion period approaches, we kindly ask if our responses have satisfactorily addressed your concerns. Your feedback would be greatly appreciated, and we would be delighted to engage in further discussions if needed.
>
> Sincerely,
>
> The Authors

---

> ### Author Response · Authors · 2024-12-01
>
> Dear Reviewer kma8
>
> We appreciate your response and the contribution your feedback has made to improving our work! As the end of the discussion period approaches, if all your queries have been addressed, we kindly ask you to consider raising your rating. If you still have any doubts or reservations about our work, we are more than willing to engage in further discussion with you.
>
> Sincerely,
>
> The Authors

---

### Official Review · Reviewer_oMQb · 2024-11-04

**Soundness:** 3
**Presentation:** 3
**Contribution:** 3
**Rating:** 6
**Confidence:** 4

**Summary:**

This paper proposes a new approach for dataset distillation by introducing a class-aware conditional mutual information (CMI) metric to address challenges in creating compact, representative synthetic datasets. Traditional dataset distillation methods often compress feature similarity without considering class-specific complexity, making it hard for models to generalize across different classes. This work leverages CMI as a regularization constraint, optimizes synthetic datasets and improves training efficiency as well as model performance.

**Strengths:**

1.	The idea of using CMI in dataset distillation to address the inherent class-aware complexity issue is interesting.
2.	The experiments are conducted based on multiple datasets and various model architectures, providing solid evidence for the method's effectiveness.
3.	The proposed method CMI is a versatile, "plug-and-play" regularization component that can be applied to numerous dataset distillation methods, such as DSA, MTT, and IDC. This flexibility allows the approach to generalize across different scenarios and highlights its robustness.
4.	By controlling the complexity of the synthetic dataset, the CMI-enhanced loss achieves faster convergence and reduces the number of required training iterations, which is particularly beneficial for large-scale datasets and resource-intensive models.

**Weaknesses:**

1.	While the paper demonstrates the CMI constraint’s benefits clearly, this method also introduces additional computation overhead, especially when dealing with high-resolution datasets. Although the authors briefly mention several strategies for mitigating this cost (e.g., reducing CMI calculations frequency), a more thorough discussion on balancing cost and performance might strengthen the practical feasibility.
2.	Although empirical evidence is strong, the theoretical basis for CMI as a regularization term could be expanded. Specifically, further details on how CMI inherently captures class complexity or why it is preferable over alternative complexity measures would provide deeper insight.
3.	While the experiments on Tiny-ImageNet and ImageNet-1K are promising, it remains unclear how the proposed method scales with even larger datasets or more complex models, such as those used in real-world applications with hundreds of classes. Additional experiments in such contexts would further show the robustness of this method.

**Questions:**

1.	The paper is well-organized and clearly presents the methodology, results, and analyses. Figures and tables are effectively used to convey improvements and insights. However, further explanation of certain key terms, such as "empirical CMI," might enhance accessibility for readers unfamiliar with the topic.
2.	The ablation studies conducted to assess the influence of the weighting parameter on the CMI constraint are informative. Still, a broader exploration of other hyperparameters affecting CMI estimation, such as the dimensionality of feature space and network depth, could reveal potential optimizations.
3.	The potential of CMI for real-world applications, such as federated learning or privacy-preserving tasks, is not discussed. Given the emphasis on dataset distillation's applications in these areas, an exploration of how CMI might support these domains would align well with the broader goals of dataset distillation research.

---

> ### Author Response · Authors · 2024-11-21
>
> We sincerely express our gratitude for dedicating your valuable time to providing insightful suggestions that can enhance our paper. Your praise regarding our methodology,
> experiments, and contributions have greatly encouraged and motivated us. Our detailed responses to all of your concerns are presented below.
>
> > **Weakness 1: Balance between training cost and performance.**
>
> Thanks for you suggestion on improving the experimental completeness, we conduct experiments on reducing the calculation frequency of CMI constraints as shown in table below. We report the performance and additional time consumption under different optimization settings, where the ratio of CMI optimization steps to distillation loss optimization steps is set to 1/10, 1/5, and 1. It can be observed that even under significantly reduced optimization step settings, the CMI constraint still achieves stable performance improvements.
> |       |         |      |     CIFAR10    |         |      |     CIFAR100    |         |
> |-------|---------|-------------|---------|---------|--------------|---------|---------|
> |       |         | 1/10 | 1/5 | 1   | 1/10 | 1/5 | 1   |
> | IDM   | additional time (s)| 0.05 | 0.1 | 0.5 | 0.5  | 2.5 | 5   |
> |       | acc (%) | 61.9±0.3 | 62.1±0.2 | **62.2±0.3** | 46.7±0.1 | 47.0±0.2 | **47.2±0.4** |
> | IDC   | additional time (s)| 0.4  | 0.8 | 4.4 | 3.2  | 6.4 | 32.0 |
> |       | acc (%) | 69.4±0.3 | 69.7±0.2 | **70.0±0.3** | 46.0±0.3 | 46.4±0.3 | **46.6±0.3** |
>
> > **Weakness 2: Comparison with other complexity measure.**
>
> Compare to other complexity metric which evaluate the complexity of a single data point (e.g., GraDN Score [1]), CMI is a class-based concept, considering all the feature vectors within one class as a whole. On the other hand, CMI enables the computation process to follow a deterministic mathematical expansion, results in a more accurate information metric. Moreover, the insight of minimizing CMI stems from constraining the complexity of the overall synthetic dataset, making it a versatile plugin. Here, we list several information metrics used in previous distillation methods and make a brief summarization:
>
> **Influence Function [2]**. influence function determines the importance of a sample by comparing the loss difference after removing a specific sample. However, this method requires retraining the model for each sample, leading to a significant computational overhead, making it impractical to optimize. Thus [3] only consider it as a evaluation metric.
>
> **Mutual Information (MI)**. MIM4DD [4] increases the MI between the synthetic and original datasets by incorporating a contrastive learning framework. Although it is not explicitly used for complexity constraints, MI is still an information metric and is thus included in our comparison.
>
> **Mean-square Error (MSE)**. IID [5] uses the MSE loss between each sample and its corresponding class center as a regularization, which is applicable only to distribution matching methods.
>
> **GraDN Score**.  SDC [6] uses the GraDN Score of the synthetic dataset corresponding to the proxy model during training as a regularization, which is applicable only to gradient matching methods.
>
> For a more intuitive comparison, we have compared the performance of our work with [4], [5], and [6] as shown in **Table 1** provided in **Sec 5.2** and **Table 8** provided in **Sec A.1**. We provide an additional comparative experiment where we remove another regularization term (i.e., MSE loss between the variance matrix of the synthetic dataset and that of the real dataset) from IID and compare it with our method in table below. It can be observed that CMI not only accelerates model training but also achieves better constraint performance across various distillation methods and datasets.
> |          | SVHN      |         | CIFAR10   |         | CIFAR100   |         |
> |----------------|-----------|---------|-----------|---------|------------|---------|
> |                | 10        | 50      | 10        | 50      | 10         | 50      |
> | IID-DM$^\dagger$  | 74.5±0.2 | 84.0±0.3 | 52.5±0.3  | 64.5±0.4 | 31.5±0.3  | 43.0±0.1 |
> | **DM+CMI**     | **77.9±0.4** | **84.9±0.4** | **52.9±0.3** | **65.8±0.3** | **32.5±0.4** | **44.9±0.2** |
> | IID-IDM$^\dagger$  | 81.8±0.1 | 84.7±0.3 | 59.4±0.2  | 68.3±0.2 | 45.8±0.4  | 51.0±0.3 |
> | **IDM+CMI**    | **84.3±0.2** | **88.9±0.2** | **62.2±0.3** | **71.3±0.2** | **47.2±0.4** | **51.9±0.3** |
>
> [1] Deep learning on a data diet: Finding important examples early in training. In NeurIPS, 2021.
>
> [2] Understanding black-box predictions via influence functions. In ICML, 2017.
>
> [3] What is Dataset Distillation Learning? In ICML, 2024.
>
> [4] MIM4DD: Mutual Information Maximization for Dataset Distillation. In NeurIPS, 2023.
>
> [5] Exploiting Inter-sample and Inter-feature Relations in Dataset Distillation. In CVPR, 2024.
>
> [6] Not All Samples Should Be Utilized Equally: Towards Understanding and Improving Dataset Distillation. In arXiv, 2024.

---

> ### Author Response · Authors · 2024-11-21
>
> > **Weakness 3: Experimental results with larger datasets and more complex models.**
>
> Thank you for your valuable suggestions regarding enhancing the robustness of the method. We conduct additional experiments on a larger dataset (i.e., ImageNet-21K) and more complex model (e.g., ResNet101) by appling CMI constraint on SRe$^2$L to evaluate the performance of our proposed method. The results shown in table below demonstrate that our method achieves significant performance improvements under all settings, validating the generality of the proposed method. We will include additional relevant experiments in the final version of the manuscript.
> | Dataset         | IPC | ResNet-18        |          | ResNet-50        |          | ResNet-101       |          |
> |------------------|-----|------------------|----------|------------------|----------|------------------|----------|
> |                  |     | SRe$^2$L | Ours         | SRe$^2$L | Ours         | SRe$^2$L | Ours         |
> |   ImageNet-1K    | 10  | 21.3±0.6         | **24.2±0.3** | 28.2±0.4         | **30.8±0.5** | 30.5±0.3         | **23.0±0.1** |
> |                  | 50  | 46.8±0.2         | **49.1±0.1** | 55.6±0.2         | **58.4±0.3** | 56.4±0.3         | **58.7±0.2** |
> |                  | 100 | 52.8±0.3         | **54.6±0.2** | 61.1±0.1         | **63.6±0.3** | 62.5±0.2         | **64.2±0.3** |
> | ImageNet-21K     | 10  | 18.5±0.4         | **21.6±0.2** | 27.2±0.3         | **29.8±0.4** | 27.5±0.1         | **30.1±0.2** |
>
> > **Question 1: Definition of empirical CMI.**
>
>
> Thank you for your suggestion on improving the presentation in the paper. Here, we propose the term **empirical CMI** to differentiate it from the CMI formula derived through explicit mathematical expressions in **Eq. 8**. In practical applications , empirical originates from the estimation of $P_{\hat{Z} \mid Y}$, i.e., using $Q_{\text{emp}}^{Y}$ obtained by summing over all sampled points and averaging as a numerical approximation of $P_{\hat{Z} \mid Y}$. For a specific synthetic dataset $\mathcal{S_y}$ = $\lbrace(s_1, y)$, $(s_2, y)$, ...,  $(s_n, y)\rbrace$ given $Y = y$,
> $Q_{\mathrm{emp}}^{y} = \frac{1}{|\mathcal{S_y}|} \sum_{s_i \in \mathcal{S_y}} P_{s_i}$.
> We then use $Q_{\mathrm{emp}}^{y}$ as the numerical estimate of $P_{\hat{Z} \mid Y}$ and substitute it into **Eq. 8** to calculate the computable CMI value.
>
> > **Question 2: Ablation study of diverse model architectures.**
>
> Thank you for your suggestion on improving the ablation experiments. In **Table 7** provided in **Sec 5.5**, we present the performance when using ConvNet and ResNet18 as proxy models for calculating CMI. Following your advice, we have conducted more comprehensive ablation experiments. Under the CIFAR10 with IPC=10, we utilized simply pre-trained ConvNet, AlexNet, ResNet18, and VGG11 as proxy models to compute CMI for both DM and DSA.
>
> We sequentially report the following metrics: the depth (i.e., number of convolution layer), the dimensionality of the feature space, and the accuracy of the simply pre-trained proxy model; the start CMI value of the synthetic dataset, the CMI value at the end of the optimization process w.o. and w. CMI constraint (cstr) of the synthetic dataset, and the accuracy of the model trained using the synthetic dataset. As shown in table below, it can be observed that the dimensionality of the feature space has a greater impact on CMI values than network depth, and CMI effectively reflects the degree of clustering in $\hat{Z}$ under the same dimensional conditions. Nonetheless, the proposed CMI constraint effectively regulates the complexity of the synthetic dataset, demonstrating the strong generalization capability of our approach.
>
> **Ablation study of proxy model architecture on DM**
> |                           |   ConvNet   |   AlexNet   |    VGG11    |   ResNet18   |
> |---------------------------|-------------|-------------|-------------|--------------|
> | **Depth**                 |      3      |      5      |      8      |      17      |
> | **Dimensionality**        |    2048     |    4096     |     512     |     512      |
> | **Model Acc (%)**         |    79.8     |    80.2     |    79.4     |    79.5      |
> | **Start CMI value**       |   0.1045    |   1.9338    |   0.0021    |   0.0016     |
> | **End CMI value w.o. cstr** |  0.0866    |   1.6217    |   0.0059    |   0.0066     |
> | **End CMI value w. cstr** |  0.0326    |   0.6642    |   0.0006    |   0.0004     |
> | **Performance (%)**       |    51.2     |    50.8     |    52.4     |    **52.9**      |

---

> ### Author Response · Authors · 2024-11-21
>
> > **Question 2: Ablation study of diverse model architectures**
>
> The results on the DSA are shown in the table below.
>
> **Ablation study of proxy model architecture on DSA**
> |                           |   ConvNet   |   AlexNet   |    VGG11    |   ResNet18   |
> |---------------------------|-------------|-------------|-------------|--------------|
> | **Depth**                 |      3      |      5      |      8      |      17      |
> | **Dimensionality**        |    2048     |    4096     |     512     |     512      |
> | **Model Acc (%)**         |    79.8     |    80.2     |    79.4     |    79.5      |
> | **Start CMI value**       |   0.1045    |   1.9338    |   0.0021    |   0.0016     |
> | **End CMI value w.o. cstr** |  0.0825    |   1.8755    |   0.0076    |   0.0051     |
> | **End CMI value w. cstr** |  0.0455    |   0.8875    |   0.0005    |   0.0004     |
> | **Performance (%)**       |    53.2     |    53.4     |    **54.8**     |    54.7      |
>
> >  **Question 3: Application on downstream tasks**
>
> Thanks for the suggestion. Following your guidance, we explore the proposed method across two downstream tasks commonly used in dataset distillation: federated learning (FL) [7] and continual learning (CL) [8]. The experimental results are presented in the table below. It can be observed that since our proposed CMI constraint effectively provides clearer classification boundaries for the synthetic datasets, thereby offering more valuable guidance information in these real-world applications.
>
> **Federated learning**
> | IPC |    DM    |  DM + CMI  |
> |-----|----------|------------|
> |  3  |   53.6   |  **54.1**  |
> |  5  |   62.2   |  **63.0**  |
> | 10  |   69.2   |  **70.4**  |
>
> **Continual learning**
> | Number of Classes |   IDC   |  IDC + CMI  |
> |-------------------|---------|-------------|
> |        20         |  65.5   |  **67.2**   |
> |        40         |  61.2   |  **62.4**   |
> |        60         |  54.5   |  **56.1**   |
> |        80         |  50.4   |  **53.6**   |
> |       100         |  46.5   |  **50.2**   |
>
> [7] FedDM: Iterative Distribution Matching for Communication-Efficient Federated Learning. In CVPR, 2023.
>
> [8]  Gdumb: A simple approach that questions our progress in continual learning. In ECCV, 2020.

---

> ### Author Response · Authors · 2024-11-24
>
> Dear Reviewer oMQb
>
> Thank you once again for dedicating your valuable time to reviewing our paper and providing constructive comments! Considering the limited time available and to save the reviewer's time, we summarize our responses here.
>
> - **1. Balance between training cost and performance:**
>   - We provide an additional experiment to reduce the frequency of CMI computation.
>
> - **2. Comparison with other complexity measure:**
>   - We enumerated various information metrics and analyzed their theoretical shortcomings.
>   - We provided a practical comparison of CMI constraint with other applicable information metrics.
>
> - **3. Experimental results with larger datasets and more complex models:**
>   - We provide additional experimental results on ImageNet-1/21K and ResNet-18/50/101.
>
> - **4. Definition of empirical CMI:**
>   - We present a more intuitive explanation for empirical CMI.
>
> - **5. Ablation study of diverse model architectures:**
>   - We employ various proxy models on different distillation methods to compute and minimize CMI.
>   - we demonstrate the generalization and robustness of the CMI constraint through experimental results.
>
> - **6. Application on downstream tasks:**
>   - We validate the effectiveness of CMI in enhancing downstream tasks in federated learning scenarios.
>   - We confirm the benefits of CMI for downstream tasks in continual learning settings.
>
>
> As the end of the discussion period approaches, we kindly ask if our responses have satisfactorily addressed your concerns. Your feedback would be greatly appreciated, and we would be delighted to engage in further discussions if needed.
>
> Sincerely,
>
> The Authors

---

### Author Response · Authors · 2024-11-21

We sincerely express our gratitude for dedicating your valuable time to providing insightful suggestions that can enhance our paper. Your praise regarding our methodology, experiments, and contributions have greatly encouraged and motivated us. Our detailed responses to all of your conerns are presented below.

Encouragingly, reviewers praise the insight of using CMI in dataset distillation ( `R#oMQb` , `R#kma8` ), solid theoretical foundation ( `R#kma8` , `R#zh5i` ), robustness ( `R#oMQb` , `R#kma8` ) and comprehensive experiments ( `R#oMQb` , `R#kma8`, `R#i6mf`, `R#zh5i` ) of our work. To better address the reviewers' concerns, we provide point-by-point responses to each reviewer below. If there are any other questions, please don't hesitate to discuss them with us.

---

### Comment · Area_Chair_cqbd · 2024-11-23
**Please review author response**

Dear reviewer,

Could you review the author response and let them know if you are satisfied with it or if you have any additional questions?

Kind regards,

Your AC

---

> ### Comment · Area_Chair_cqbd · 2024-11-29
>
> Dear Reviewers,
>
> If you haven’t already, could you kindly review the author response and let the authors know if you are satisfied with it or if you have any additional questions? Your contribution is greatly appreciated.
>
> Thank you!
>
> Kind regards,
>
> Your AC

---

### Author Response · Authors · 2024-11-25

Thanks all the reviewers once again for dedicating your valuable time to reviewing our paper and providing constructive comments!

As the end of the discussion period approaches, we kindly ask if our responses have satisfactorily addressed your concerns. Your feedback would be greatly appreciated, and we would be delighted to engage in further discussions if needed.

Sincerely,

The Authors

---

> ### Comment · Area_Chair_cqbd · 2024-11-26
>
> Dear reviewers,
>
> Thank you if you have already reviewed the author response.
>
> If not, could you kindly review it and let the authors know if you are satisfied with their response or if you have any further questions?
>
> Kind regards,
>
> Your AC

---

### Author Response · Authors · 2024-12-02

Dear Reviewers,

Thanks again for providing the constructive review. We have tried our best to address your concerns point by point so that you can have a better understanding of our work.

However, since the discussion period is going to end in one day, we really want to hear from you and see if we have addressed your concerns. Please participate in the discussion because your feedback is important for us to improve our work. If all your queries have been addressed, we kindly ask you to consider raising your rating. If you still have other questions, we are more than willing to explain in detail.

Looking forward to your reply.

Sincerely,

The Authors

---

### Meta-Review · Area_Chair_cqbd · 2024-12-24

**Metareview:**

This work focuses on the complexity of the synthetic dataset in dataset distillation. It proposes to apply conditional mutual information (CMI) to assess this complexity and uses this criterion as a regulariser of the existing dataset distillation methods. Experimental study shows that the proposed regulariser can be used as a plug-and-play component to work with various distillation losses to achieve improved performance and faster convergence.

Reviewers comment that the proposed method is interesting, versatile and effective; the experiments are comprehensive and solid; this work is innovative and makes significant contribution; the review is thorough; and it is technically sound. Meanwhile, the reviewers raise the issues related to the extra computational overhead, the theoretical basis of using CMI as a regulariser, the scalability with respect to dataset size and model complexity, experimenting with new methods, the novelty of this work, the discussion of limitation, the difference from an existing work on CMI, and the lack of explanation.

The authors provide a response to each of the raised issues by further clarification, additional experiments, and pseudo code. The rebuttal is overall detailed and effective. Reviewer oMQb explicitly replies that the rebuttal addresses most of the concerns they raised. However, Reviewer Zh5i still has concerns related to the difference of this work from an existing work on CMI and therefore the adequacy of the novelty. The final ratings are 6, 6, 6 and 3 (Reviewer Zh5i).

AC carefully reviews the submission, the comments, the rebuttals, the discussion, and the message from the authors. AC generally concurs with the reviewers' comments on this work. In addition, although this work adopts the CMI criterion from the literature and therefore does not have significant contribution in this regard, it does provide new insights on the complexity of the synthetic dataset and innovatively utilises CMI to consider this information in the optimisation of dataset distillation. Also, experimental study verifies the effectiveness and the generality of this work. In the follow-up discussion between reviewers and AC, Reviewer Zh5i indicates that they will not oppose acceptance if the majority of reviewers strongly advocate for it.

Taking all the factors into account, AC would like to recommend this work for acceptance. Meanwhile, the authors are strongly suggested to further improve the clarity of this work, provide more explanation, and explicitly list the differences from the work raised in the comments of Reviewer Zh5i.

**Additional Comments On Reviewer Discussion:**

The reviewers raise the issues related to the extra computational overhead, the theoretical basis of using CMI as a regulariser, the scalability with respect to dataset size and model complexity, experimenting with new methods, the novelty of this work, the discussion of limitation, the difference from an existing work on CMI, and the lack of explanation. The authors provide a response to each of the raised issues by further clarification, additional experiments, and pseudo code. The rebuttal is overall detailed and effective. Reviewer oMQb explicitly replies that the rebuttal addresses most of the concerns they raised. However, Reviewer Zh5i still has concerns related to the difference of this work from an existing work on CMI and therefore the adequacy of the novelty. The final ratings are 6, 6, 6 and 3 (Reviewer Zh5i).

AC carefully reviews the submission, the comments, the rebuttals, the discussion, and the message from the authors. AC generally concurs with the reviewers' comments on this work. In addition, although this work adopts the CMI criterion from the literature and therefore does not have significant contribution in this regard, it does provide new insights on the complexity of the synthetic dataset and innovatively utilises CMI to consider this information in the optimisation of dataset distillation. Also, experimental study verifies the effectiveness and the generality of this work. In the follow-up discussion between reviewers and AC, Reviewer Zh5i indicates that they will not oppose acceptance if the majority of reviewers strongly advocate for it. Taking all the factors into account, AC would like to recommend this work for acceptance.

---

### Decision · Program_Chairs · 2025-01-22

Accept (Poster)